# Border-associated macrophages mediate the neuroinflammatory response in an alpha-synuclein model of Parkinson disease

A. M. Schonhoff [1,2], D. A. Figge[3], G. P. Williams [1,2], A. Jurkuvenaite [1,2], N. J. Gallups [2], G. M. Childers [2], J. M. Webster [1,2], D. G. Standaert [1,2], J. E. Goldman [1,4] & A. S. Harms [1,2] ✉

Dopaminergic cell loss due to the accumulation of α-syn is a core feature of the pathogenesis of Parkinson disease. Neuroinflammation specifically induced by α-synuclein has been shown to exacerbate neurodegeneration, yet the role of central nervous system (CNS) resident macrophages in this process remains unclear. We found that a specific subset of CNS resident macrophages, border-associated macrophages (BAMs), play an essential role in mediating α-synuclein related neuroinflammation due to their unique role as the antigen presenting cells necessary to initiate a CD4 T cell response whereas the loss of MHCII antigen presentation on microglia had no effect on neuroinflammation. Furthermore, α-synuclein expression led to an expansion in border-associated macrophage numbers and a unique damage-associated activation state. Through a combinatorial approach of single-cell RNA sequencing and depletion experiments, we found that border-associated macrophages played an essential role in immune cell recruitment, infiltration, and antigen presentation. Furthermore, border-associated macrophages were identified in postmortem PD brain in close proximity to T cells. These results point to a role for border-associated macrophages in mediating the pathogenesis of Parkinson disease through their role in the orchestration of the α-synuclein-mediated neuroinflammatory response.

Parkinson disease (PD) is the most common neurodegenerative movement disorder, characterized pathologically by the abnormal accumulation of alpha-synuclein (α-syn) in Lewy bodies and neurites and resulting in the loss of dopamine-producing neurons in the substantia nigra pars compacta (SNpc). Due to activation of tissue-resident macrophages and the infiltration of both innate and adaptive immune cells, neuroinflammatory mechanisms have been strongly implicated in the neurodegeneration associated with α-syn accumulation[1,2]. Polymorphisms in the HLA-DR (MHCII)

locus and chronic gut inflammation both increase the risk of developing PD[3], while treatment with anti-Tumor Necrosis Factor and nonsteroidal anti-inflammatory drugs reduce PD risk[4]. In postmortem tissue, Lewy pathology is accompanied by the enhanced expression of HLA-DR (MHCII) on microglia, infiltration of T cells into the brain, and neuronal cell death[5]. Neuroimaging studies in humans have confirmed chronic myeloid activation in the brain of PD patients, and α-syn-reactive T cells have been found circulating in the blood of patients[6,7].

[1]Aligning Science Across Parkinson's (ASAP) Collaborative Research Network, Chevy Chase, MD, USA. [2]Center for Neurodegeneration and Experimental Therapeutics, Department of Neurology, The University of Alabama at Birmingham, Birmingham, AL 35294, USA. [3]Department of Pathology, School of Medicine, University of Alabama at Birmingham, Birmingham, AL, USA. [4]Department of Pathology and Cell Biology, Columbia University, New York, NY 10032, USA. ✉e-mail: anharms@uab.edu

Many of the inflammatory features seen in human disease are similarly observed in transgenic and viral α-syn overexpression mouse models of PD, with microgliosis, T cell infiltration, and entry of peripheral monocytes occurring prior to neurodegeneration[8,9]. In a viral overexpression model, global knockouts of MHCII, the Class II transcriptional co-activator (CIITA), or of CD4 were found to be neuroprotective, indicating a pivotal role for communication between the innate and adaptive immune systems in α-syn-driven neurodegeneration[10,11]. Surprisingly, knockout of CD8+ T cells had no effect on infiltration of other peripheral immune cells and neurodegeneration[10], suggesting that antigen presentation specifically to CD4+ T cells is a critical mechanism of PD pathogenesis. A key question remains: which cells are responsible for the antigen presentation that is driving the α-syn-induced neurodegeneration? Previously we have shown that local antigen presentation via MHCII expression is essential to the loss of dopaminergic neurons in the SNpc[11], indicating that CNS resident macrophages (CRMs) may be key antigen-presenting cells that orchestrate neuroinflammatory and neurodegenerative responses.

CRMs include microglia in the parenchyma and border-associated macrophages (BAMs) that reside in the choroid plexus, dural and subdural meninges, and are adjacent to the vasculature within the perivascular space[12]. It is important to note that BAMs have also been referred to as CNS-associated macrophages (CAMs) to highlight their anatomical location[12]. Both CRM populations arise from early erythromyeloid progenitor cells in the fetal yolk sac but segregate by day E10.5 into two distinct populations that comprise the majority of CNS immune cells[12–14]. Recent work in human and animal models of neurodegeneration have identified unique activation states of microglia[15,16], and it is thought that these cell states likely play protective or harmful roles depending on their context[17]. Extensive single-cell RNA sequencing studies have identified expanded populations of activated microglia in aged mice and in the context of Alzheimer's disease[18]. However, whether these disease-associated activation states are seen in PD and what their functional relevance remains unknown. While the role of BAMs in neurodegenerative disease remains unknown, they have been shown to support blood vessel repair, mediate blood–brain barrier permeability, produce reactive oxygen species, secrete chemokines to orchestrate monocyte and granulocyte recruitment, and are highly phagocytic, indicating they may play a role in disease pathogenesis as effectors or regulators of immune responses[12,19–21].

To further understand the role of CRMs in PD, we sought to determine which cells are responsible for antigen presentation in PD and how CRMs contribute to α-syn-mediated neuroinflammation and neurodegeneration. Utilizing an α-syn overexpression in vivo model combined with single-cell profiling technology, we have demonstrated that BAMs, not microglia, are responsible for CD4+ T cell antigen recruitment and restimulation necessary for α-syn-mediated neuroinflammation. These findings change our classical understanding of neuroinflammatory mechanisms in neurodegenerative disease, as they implicate unique and non-redundant functions for BAMs in their role of immune cell recruitment, class II antigen presentation, and T cell infiltration.

## Results

### CRM expression of MHCII is required for α-syn induced neurodegeneration in mice

Previous studies have identified a pivotal role for MHCII in the neuroinflammation and neurodegeneration associated with PD; however, the specific cells presenting antigen have remained elusive. While our work and others have highlighted an essential role for brain localized antigen presentation in α-syn induced neurodegeneration, recent evidence has shown that infiltrating monocytes are also capable of upregulating MHCII upon CNS entry. Due to this discrepancy, we

became interested in what role CRMs specifically play in the immune response to α-syn overexpression. To evaluate the roles of CRMs in PD, we used a previously published rAAV (referred to here as AAV2-GFP or AAV2-SYN) model to drive α-syn overexpression in dopaminergic neurons of the substantia nigra pars compacta. This model induces numerous parallels to human disease, including the aggregation of phosphorylated, triton insoluble α-syn, and the loss of ~25–30% of TH+ neurons in the SNpc at 6 months post-transduction[22,23]. To evaluate the importance of CRMs in α-syn induced inflammation and neurodegeneration, we conditionally deleted MHCII from CRMs using CX3CR1^CreERT2/+ IAB^fl/fl mice prior to α-syn overexpression (Fig. 1a). Tamoxifen administration into CX3CR1^CreERT2/+ IAB^fl/fl mice leads to a selective and persistent recombination in the long-lived CRMs (Fig. 1b, Supplemental Fig. 1a, b). Six weeks post-tamoxifen, allowing for turnover of monocytes and dendritic cells (Supplemental Fig. 1a, b), mice were then transduced with an adeno-associated virus that causes the overexpression of human α-syn (AAV-SYN) to selectively overexpress α-syn in nigral neurons, inducing inflammation-dependent neurodegeneration[10,23]. Four weeks post-transduction and during peak inflammatory response, we found that the selective loss of MHCII in CRMs resulted in a reduction of multiple neuroinflammatory markers including microglial activation and CD4+ T cell infiltration, with minimal effect on α-syn expression and CD8+ T cell infiltration (Fig. 1c–e, Supplemental Fig. 1c, d). Both genotypes exhibited equivalent phospho-serine 129 α-syn (pSer) pathology at 4 weeks post-transduction (Supplemental Fig. 1e, f). Six months post-transduction, depletion of MHCII from CRMs was neuroprotective against dopaminergic cell loss in the SNpc, whereas our control mice still displayed the expected ~20% loss of dopaminergic neurons found in this model of PD (Fig. 1f, g, Supplemental Fig. 1g)[23]. These results indicate that antigen presentation specifically by CRMs are essential for α-syn-induced initiation of the inflammatory response and subsequent neurodegeneration induced by α-syn.

### In vivo α-syn overexpression induces disease-associated microglia

Recent studies using single-cell RNA seq (scRNAseq) technologies on CX3CR1+ CRMs have identified two unique populations, microglia and BAMs, and have reported extensive transcriptional diversity within these subpopulations. To understand the cell type-specific transcriptional changes underlying the α-syn-driven inflammatory response, we isolated CX3CR1+ cells using FACs at four weeks post transduction with AAV2-GFP or AAV2-SYN and performed scRNAseq using the 10x Genomics platform (Fig. 2a). Using Seurat, we integrated these datasets together to specifically identify what effect α-syn overexpression had on CRMs. As expected based upon previous studies, the profiled cells separated into two distinct populations (Fig. 2b), with the majority of cells expressing classical microglial markers including *Crybb1*, *Sall1*, *Tmem119*, and *Fcrls* (Fig. 2c, d, Supplemental Fig. 2a) while the other (Cluster 8) was found to express several markers associated with BAMs such as *Clec12a*, *Ms4a7*, *Mrc1*, *Lgals3*, *CD163*, and *Pf4* (Fig. 2c, Supplemental Fig. 2a)[16,18,24]. These two populations were transcriptionally distinct (Supplemental Fig. 2b). The microglia were further subdivided into 7 unique clusters with the largest clusters representing quiescent microglia based upon their high expression of homeostatic and microglial identity genes, especially *Sall1*, *Fcrls*, *Slc2a5* (Fig. 2e). Comparing the unique cluster defining genes in our data to other previously published sets, we identified clusters 3, 4, and 5 as proliferative, activated, and interferon-responsive microglia respectively (Fig. 1d, e)[18,21]. While all microglial sub-clusters were present in both AAV2-GFP and AAV2-SYN experimental conditions, α-syn overexpression induced the expansion of clusters 4, 6, and 7 (Fig. 2f, g, Supplemental Fig. 2c, d). Interestingly, clusters 6 and 7 were defined by their expression of *Cst7*, *Lpl*, and *Apoe*, genes associated with the recently identified disease-associated microglia (DAMs) found in

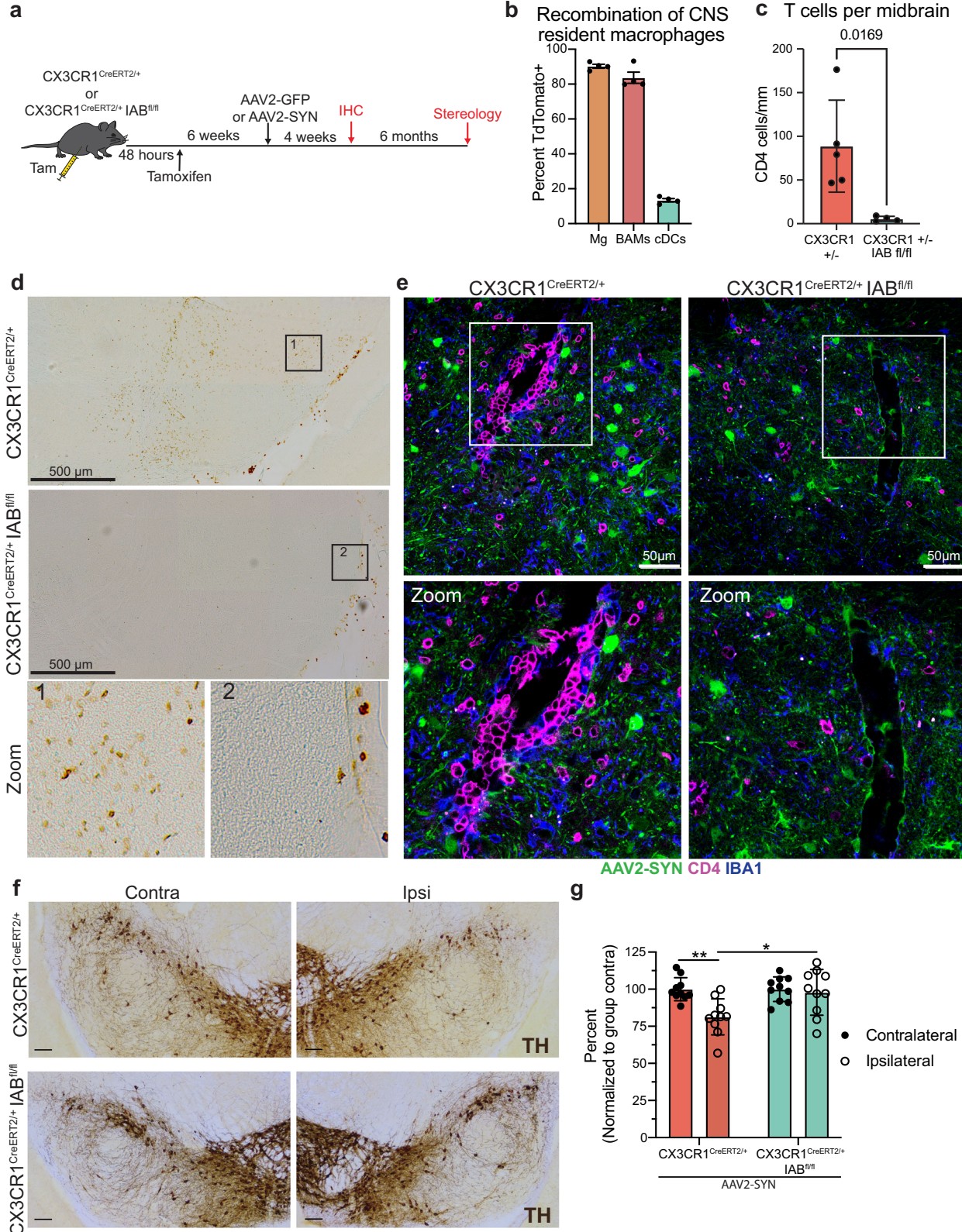

**f**

AAV2-SYN **AAV2-SYN** **CD4** **IBA1**

Alzheimer's disease (Fig. 2d, e)[15]. Using Monocle3 for pseudotime analysis, we found that the two DAM clusters likely represent an "early" and "late" activation subset (Supplemental Fig. 2e). Gene ontology of the DAM cluster defining genes revealed an enrichment for pathways involved in toll-like receptor (TLR) signaling, phagocytosis, antigen presentation and processing, and cytokine/chemokine signaling (Fig. 2h), consistent with α-syn's known role in activating TLRs on

microglia[25]. We sought to confirm our scRNA-seq data using flow cytometry on CRMs isolated from the ventral midbrains of AAV2-SYN transduced mice. Corroborating our sequencing results, we found α-syn expression had no effect on the total number of microglia but instead led to the enhanced expression of several genes found in the DAM clusters including MHCII and PD-L1 (also known as CD274) (Fig. 2i, j, Supplemental Fig. 2f). Microglia increased

**Fig. 1 | CNS resident macrophage antigen presentation mediates neurodegeneration. a** Experimental paradigm for conditional Iab deletion from CX3CR1⁺ cells. Mice received tamoxifen and 4 weeks later were given AAV2-SYN. Four weeks post AAV2-SYN, flow cytometry and immunohistochemistry were performed. Six months post-AAV, unbiased stereology was performed to quantify neurodegeneration. **b** Quantification of recombination efficiency of CNS resident macrophages, including microglia (Mg) and border-associated macrophages (BAMs), or meningeal classical dendritic cells (cDCs) 6 weeks after tamoxifen treatment using the CX3CR1^CreERT2 mice crossed to TdTomato^fl/fl reporter mice. *N* = 4 mice, unpaired two-tailed T test. *p = 0.0169. Mean ± SEM is displayed. Representative results of two independent experiments are shown. **c** Quantification of ventral midbrain CD4⁺ T cells in (**d**). *n* = 5 CX3CR1+/− mice and 4 CX3CR1+/− Iab fl/fl mice, **p < 0.01. Mean ± SD is shown. **d** Representative scans of the ventral midbrain containing the SNpc of DAB staining for CD4⁺ T cells in CX3CR1^CreERT2/+ Iab^fl/fl or CX3CR1^CreERT2/+ mice. Zoomed images are below showing higher magnification of the boxed area. Numbers indicate if the zoomed image was from CX3CR1^CreERT2/+ or CX3CR1^CreERT2/+

Iab^fl/fl mice. Representative images from two independent experiments are shown. **e** Immunofluorescent images of CD4⁺ T cell infiltration in CX3CR1^CreERT2/+ Iab^fl/fl or CX3CR1^CreERT2/+ mice. Brains are labeled with AAV2-SYN (green), CD4 (red) and IBA1 (blue). Images are taken at ×40, scale bar is 50 μm (top) or images are digitally zoomed (bottom). Representative images from two independent experiments are shown. **f** Representative DAB images of the substantia nigra pars compacta in CX3CR1^CreERT2/+ Iab^fl/fl or CX3CR1^CreERT2/+ mice. Images are labeled with tyrosine hydroxylase (TH), and both the contralateral uninjected (left) and ipsilateral injected (right) sides of the brain are shown. Scale bar is 100 μm. Representative image from one experiment is shown. **g** Quantification of unbiased stereology of TH+ immunostained neurons in the SNpc of CX3CR1^CreERT2/+Iab^fl/fl or CX3CR1^CreERT2/+ control mice who received AAV2-SYN. Quantification is performed at 6 months post AAV transduction and neuron counts are normalized to the average of the group contralateral side. Two-way ANOVA, with Bonferonni multiple comparisons correction, *n* = 10 mice per group. **p = 0.0053, *p = 0.0105. Mean ± SD is shown. *Source data is provided in the "Source data" file.

expression of Arg1 but did not change in iNOS or CD68 expression (Supplemental Fig. 2f, g). Together, these data indicate that α-syn overexpression leads to substantial changes in microglial transcriptional behavior and function contributing to the inflammatory mechanisms underlying neuroinflammation and neurodegeneration.

## Microglia are not primary CNS antigen-presenting cells in an α-syn overexpressing mouse

To specifically evaluate the importance of microglial antigen presentation in α-syn induced neuroinflammation, we selectively deleted MHCII in microglia using TMEM119^CreERT2/+ Iab^fl/fl mice. Initially, we utilized a fate-mapping mouse to test recombination upon tamoxifen administration. We found that a portion of BAMs expressed TdTomato (Fig. 3a). We therefore decided to functionally test the recombination in TMEM119^CreERT2/+ Iab^fl/fl mice. As microglia do not express high levels of MHCII at baseline, we initially confirmed microglial specificity by administering intra-nigral interferon gamma (IFNγ) 4 weeks post-tamoxifen administration to induce MHCII expression (Fig. 3b, c, Supplemental Fig. 3a). We found that 93% of Tmem119^CreERT2/+ microglia, whereas only 34% of TMEM119^CreERT2/+ Iab^fl/fl microglia expressed MHCII upon IFNγ administration (Fig. 3b, Supplemental Fig. 3b). In contrast, we found minimal deficits in BAM ability to express MHCII after IFNγ (Fig. 3c, Supplemental Fig. 3a). We then overexpressed α-syn in TMEM119^CreERT2/+ Iab^fl/fl mice and controls via AAV2-SYN to assess the role of microglial antigen presentation while preserving BAM MHCII expression (Fig. 3d). Both genotypes expressed equivalent pSer129 α-syn pathology at 4 weeks post-transduction (Supplemental Fig. 3b, c). As expected, we found no upregulation of MHCII on microglia following α-syn overexpression in the TMEM119^CreERT2/+ Iab^fl/fl mice (Fig. 3e). To our surprise, we found no change in neuroinflammation including infiltration of Ly6C^hi monocytes, CD4⁺ or CD8⁺ T cells (Fig. 3f, j). These data, combined with our findings in the CX3CR1^CreERT2/+ IAB^fl/fl mice, indicated that a non-microglial CRM is necessary for the α-syn induced neuroinflammation, specifically antigen presentation and the infiltration of peripheral immune cells.

## α-syn induces disease-associated subsets of BAMs in vivo

Given our observation that microglia are not the primary cells necessary for antigen presentation, we sought to focus on the alternative population of CRMs, BAMs, as they express MHCII at baseline and upregulate expression in other neuroinflammatory conditions[21,24]. To further characterize the transcriptional changes specific to BAMs, we isolated these cells from the larger dataset for re-analysis, identifying 8 distinct clusters that largely mirrored the heterogeneity identified in microglia (Fig. 4a). All BAMs were found to express several previously described cell defining genes, including *Mrc1*, *Ms4a7*, *Lgals3*, and *Pf4* (Fig. 4c, Supplemental 4a) albeit with varying expression between clusters[26]. Similar to microglia, the majority of BAMs were quiescent

and highly expressed identity genes (Fig. 4b, c), with additional subsets that were proliferating, interferon-responsive, or undergoing early stages of activation (clusters 3-5) (Fig. 4a–c). Interestingly, our scRNAseq data established that several cluster defining genes for the BAMs included many disease-associated microglia (DAM)-defining genes, including *Apoe*, *Clec7a*, *Itgax* (CD11c), class II-related genes (*H2-Aa*, *Cd74*), and *Cd274* (the gene for PD-L1) (Fig. 4a–c). Most clusters, particularly clusters 6 and 7, also expressed several genes involved in phagocytosis (*Cd68*), whereas activated clusters expressed genes involved in inflammation (*Il1b*), lymphocyte chemotaxis (*Ccl5*, *Cxcl10*), and remodeling of the extracellular matrix (*Mmp14*) (Fig. 4b, c, Supplemental Fig. 4a). Furthermore, they most highly expressed *Gpnmb*, a gene that has been genetically implicated in PD and had high expression of several DAM defining genes such as *Apoe*, *Lgals3*, and *Fabp5* (Fig. 4b, c)[24,27–29]. As previously seen in the DAMs, pseudotime analysis using Monocle3 suggested that these two clusters likely represented differential stages of activation (Supplemental Fig. 4b). We therefore named these two clusters "disease-activated BAMs" (DaBAMs), as they also resembled previously described populations[26]. KEGG pathway analysis of the cluster defining genes for DaBAMs similarly revealed an enrichment for several key inflammatory processes including cytokine signaling, antigen processing, and antigen presentation (Supplemental Fig. 4c). Prominent changes in both total BAMs and in cluster distributions occurred following α-syn expression, with particularly significant AAV-SYN dependent expansion of the two DaBAM clusters (BAM clusters 6 and 7) (Fig. 4d–e, Supplemental Fig. 4d). We validated this increase in total subdural BAMs using flow cytometry and immunohistochemistry (Fig. 4f, Supplemental Fig. 5a), and confirmed that α-syn overexpression led to even higher expression of MHCII on BAMs than microglia (Fig. 4g), along with the increased expression of key proteins involved in T cell interactions including PD-L1 (Cd274), and CD80 (Fig. 4h–i, Supplemental Fig. 4f), similar to what has been found in other neuroinflammatory conditions[21]. Using immunohistochemistry (IHC), we validated the α-syn-induced effects on BAMs found in our scRNAseq results, noting expression of Apoe⁺, GPNMB⁺, and CD68⁺ macrophages in the perivascular space and subdural meninges (Fig. 4j, Supplemental Fig. 4e). The overall increase in BAM number was driven by both local proliferation of CX3CR1⁺ BAMs as well as engraftment of peripherally-derived cells (Supplemental Fig. 5b–e). Collectively, these findings suggest BAMs play an essential role in α-syn-induced neuroinflammatory processes by facilitating T cell recruitment, antigen restimulation, and CNS parenchymal entry.

## BAMs are required for α-synuclein-mediated neuroinflammation in mice

To directly examine the importance of BAMs to α-syn-mediated neuroinflammation, we specifically depleted BAMs using clodronate-filled liposomes (CL) (Fig. 5a). CL selectively depleted both

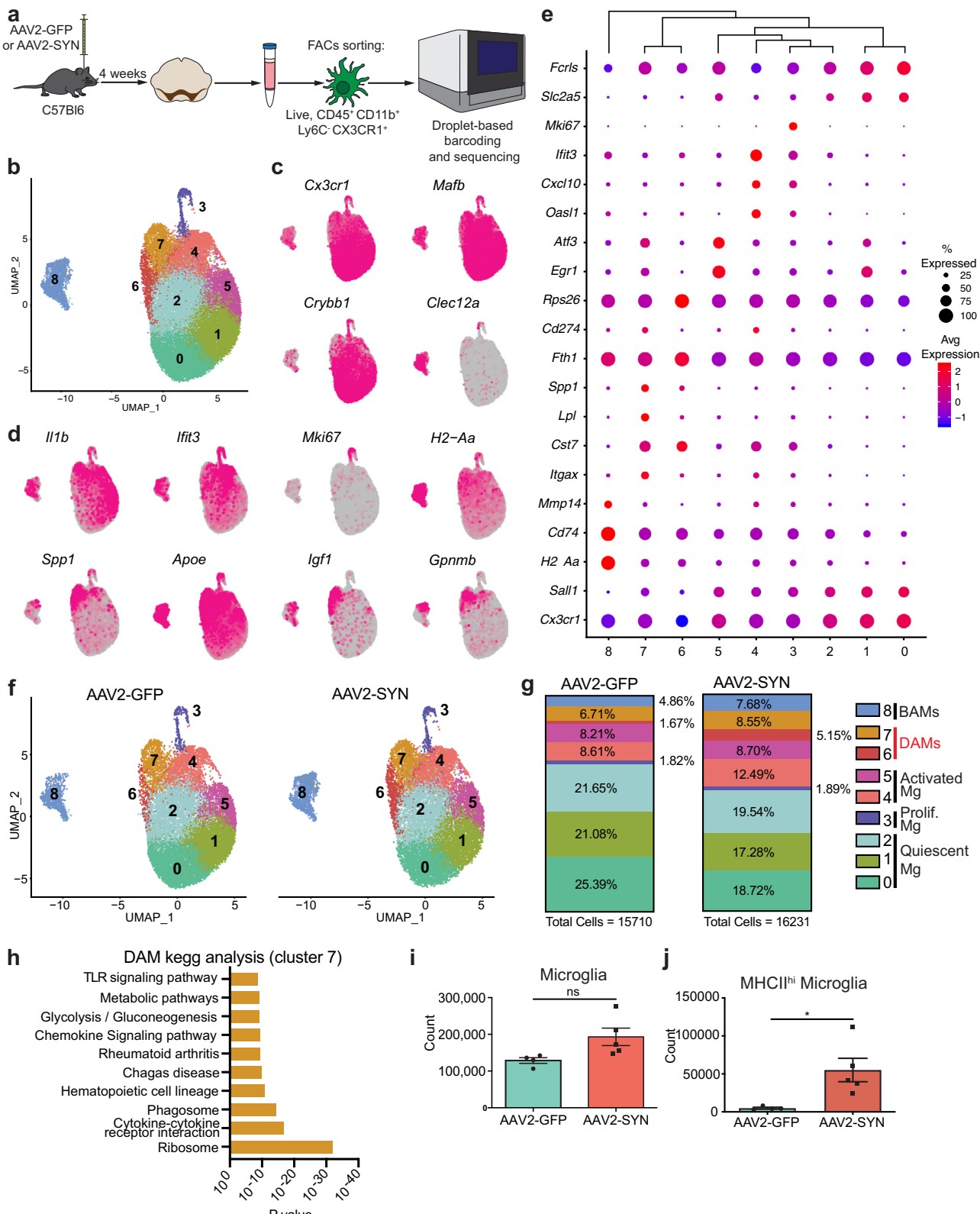

perivascular/subdural BAMs (27% reduction in perivascular BAMs at baseline) and dural BAMs (74% reduction at baseline) with no effect on the numbers of microglia, circulating monocytes, or meningeal dendritic cell (cDC) populations (Fig. 5b, c, Supplemental Fig. 6b–d). While the reduction of total perivascular BAMs was modest, the percentage of MHCII+ perivascular BAMs was significantly decreased (29.7% in SL reduced to 8.1% in CL) (Supplemental Fig. 6b). CL-treated animals

expressed substantially less MHCII in the SNpc following α-syn over-expression on cells that morphologically resembled both microglia and infiltrating myeloid cells (Fig. 5d). Furthermore, BAM depletion recapitulated our findings in CX3CR1[CreERT2/+] IAB[fl/fl] mice, reducing multiple measures of neuroinflammation including microglial activa-tion, and the number of infiltrating Ly6C[hi] monocytes and CD4[+] T helper cells (Supplemental Fig. 6e, f, Fig. 5f–j). Together, these findings

**Fig. 2 | Alpha-synuclein-specific changes in disease-associated microglia.**
**a** Experimental paradigm for mononuclear cell isolation and single-cell RNA sequencing of brain-resident macrophages, defined by CD45$^+$ CD11b$^+$ Ly6C$^-$ CX3CR1$^+$. Mice were injected with AAV2-SYN or AAV2-GFP into the SNpc, and cells were isolated 4 weeks later. **b** Integrated U-MAP projection of 31,941 cells. In total, eight microglia clusters and one BAM cluster were identified. $n = 4$ samples, with 3–4 ventral midbrains pooled per sample. **c** Integrated U-MAP plots colored for expression of microglial and border-associated macrophage identity genes **d** Integrated U-MAP plots colored for expression of cluster-defining genes. **e** Dot plot corresponding to the integrated U-MAP plot of brain-resident macrophages demonstrating cluster-identifying genes. Dot size represents the percentage cluster cells expressing the gene and dot color represents its average expression within the cluster. **f** U-MAP projections of brain-resident macrophage clusters from AAV2-GFP or AAV2-SYN midbrains. $N = 2$ samples per group, with 3–4 ventral midbrains pooled per sample. **g** Quantification of relative abundance for each cluster in response to AAV2-GFP or AAV2-SYN. Percentage of total population by each cluster

is displayed on graph. Cells are labeled according to their genetic profiles. **h** KEGG analysis of DAMs (cluster 7) displaying enrichment for processes such as TLR signaling pathways, chemokine signaling, cytokine-cytokine receptor interactions, and phagosome. The Webgestalt online tool with hypergeometric testing and a Benjamini Hochberg correction for multiple tests was used. The top 10 pathways with the most significant $p$ values (under $p = 0.05$) and 2 or more genes in the group were identified and displayed. **i** Quantification of flow cytometric data. Microglia were gated as live, CD45$^+$ CD11b$^+$ CX3CR1$^+$ Ly6C$^-$ and CD38$^-$. Unpaired T-test, two-tailed, $n = 4$ AAV2-GFP and 5 AAV2-SYN transduced samples per group with 2 pooled ventral midbrains per sample. Mean ± SD is shown. Representative results of three independent experiments are shown. **j** Quantification of flow cytometric data investigating MHCII$^{hi}$ microglia. Unpaired T-test, two-tailed, $n = 4$ AAV2-GFP and 5 AAV2-SYN transduced samples per group with 2 pooled ventral midbrains per sample. *$p = 0.0239$. Mean ± SD is shown. Representative results of three independent experiments are shown. *Source data is provided in the "Source data" file.

---

indicate that BAMs, not microglia are required for peripheral immune cell recruitment into the brain parenchyma and antigen restimulation, a process responsible for α-syn-induced neurodegeneration.

### BAMs and T cells interact in human PD postmortem brain

BAMs in the rodent brain have been sparsely studied, and therefore their presence and role in human disease remains almost a complete mystery. We found abundant T cells and BAMs within the perivascular space of AAV2-SYN transduced animals, and often observed the two cells in close proximity, suggesting an ongoing interaction (Fig. 6a). We sought to identify correlates to these findings in human PD. However, because little is known about human BAMs, we used a combination of known markers and location to identify BAMs in midbrains of PD and, age-matched controls. We obtained human samples from the New York Brain Bank and selected both men and women aged 72–92 years of age. The PD group was defined by the presence of Lewy pathology and a clinical diagnosis of PD (Supplementary Table 1). Neurological controls had no history of PD and no Lewy pathology (Supplementary Table 1). We excluded one patient with incidental Lewy bodies but no reported symptoms of PD. The sections of ventral midbrain were double-labeled with CD68 and CD3 to visualize BAMs and T cells in the perivascular spaces around vessels penetrating the ventral midbrain and the nigra. BAMs were identified as elongated CD68$^+$ cells immediately adjacent to the parenchyma but distinctly within the perivascular space. Both neurological controls and PD midbrains contained abundant BAMs (Fig. 6b). We quantified total numbers of CD3$^+$ T cells in the ventral area of midbrains (see Methods), and the percentages of total CD3$^+$ cells that lay adjacent to CD68$^+$ BAMs within the perivascular space (Fig. 6c–e). Intriguingly, PD patients had a significantly higher percentage of the total CD3$^+$ T cell population adjacent to BAMs than did the healthy controls (Fig. 6f, g). We then sought to determine whether the CD3$^+$ T cells were CD8$^+$ or CD4$^+$. We found both cell types present in the perivascular space in close proximity to CD68$^+$ BAMs (Fig. 6h, i), mirroring what we see in our mouse model of PD, indicating a disease-associated interaction similar to that observed in mice.

### Discussion

While there is growing evidence that neuroinflammation is essential to PD pathogenesis, the underlying role of CRMs in directing this response has remained a mystery. Due to the use of transient and overlapping markers, it has been difficult to define the unique functions of individual populations of CRMs relative to infiltrating monocytes and macrophages. Utilizing a complementary combination of molecular, cellular, and pharmacological approaches, the present study defines an essential role for antigen presentation by CRMs, particularly BAMs, in the initiation of α-syn-mediated neuroinflammatory responses via peripheral immune cell recruitment and

antigen restimulation. These results point to a critical role for BAMs in neuroinflammatory responses and highlight the need for disease-modifying therapeutic targeting of BAMs in neurodegenerative disease.

In this study, we found that α-syn induces a disease-associated activation state of CRMs (DAMs and DaBAMs) similar to that previously identified in Alzheimer's disease (AD) and Amyotrophic lateral sclerosis (ALS)[15]. These disease-associated CRM subsets (DAMs and DaBAMs) were found to have decreased expression of genes defining microglial identity (*Sall1*, *Fcrls*, *Slc2a5*), with a corresponding upregulation of numerous involved in cytokine signaling (*Ccl5*, *Cxcl14*, *Ccl6*) and essential to phagocytosis (*Cd68*) (Figs. 2e and 4c). These data are consistent with α-syn's known interaction with TLRs on CRMs to initiate inflammatory cascades. We have identified a unique function of BAMs, showing that they are responsible for providing restimulation of CD4$^+$ T cells necessary for α-syn-mediated local cytokine production and parenchymal entry. Accordingly, DaBAMs were found to highly express several genes involved in T cell recruitment (*Ccl5* and *Ccl10*), antigen processing and presentation (*H2-Aa*, *Cd74*, and *Cd274*), and remodeling of the extracellular matrix (*Mmp14*) (Fig. 4c) suggesting critical roles in mediating BBB permeability, T cell restimulation, and immune cell entry. These data further corroborate other studies highlighting crucial roles of BAMs in chemokine secretion and neuroinflammation[21]. Confirming the important role BAMs play in the immune response to α-syn, depletion of BAMs reduced multiple markers of neuroinflammation including CRM activation and the entry of peripheral immune cells, while conditional knockout of MHCII on microglia had no effect on α-syn-induced neuroinflammation (Fig. 3). With their unique location, ability to initiate lymphocyte chemotaxis, and role in the remodeling of the parenchymal extracellular matrix, these data place BAMs as central mediators between the peripheral and tissue-resident immune systems and as vital to the neuroinflammatory processes in PD.

While previous research on the role of macrophages in multiple models of neurodegeneration has highlighted the importance of microglia in protein clearance and phagocytosis, there has been minimal focus on the roles of alternative subsets of CRMs. Recent studies on CRMs in AD and ALS have identified the importance of DAMs based upon their unique transcriptional and protein profiles using single cell-based technologies. Alternatively, disease-associated states of BAMs have been described in multiple studies, but their role as a functional and critical antigen-presenting cell in neuroinflammation has not been defined. Similar to AD and ALS, our scRNA seq data found a similar population of disease-associated BAMs, DaBAMs, and confirmed BAMs essential role in antigen presentation and T cell infiltration to the CNS[16,18,24]. As most of the previous work profiling macrophages in neurodegeneration has primarily focused on broad populations of myeloid cells, the present study highlights the

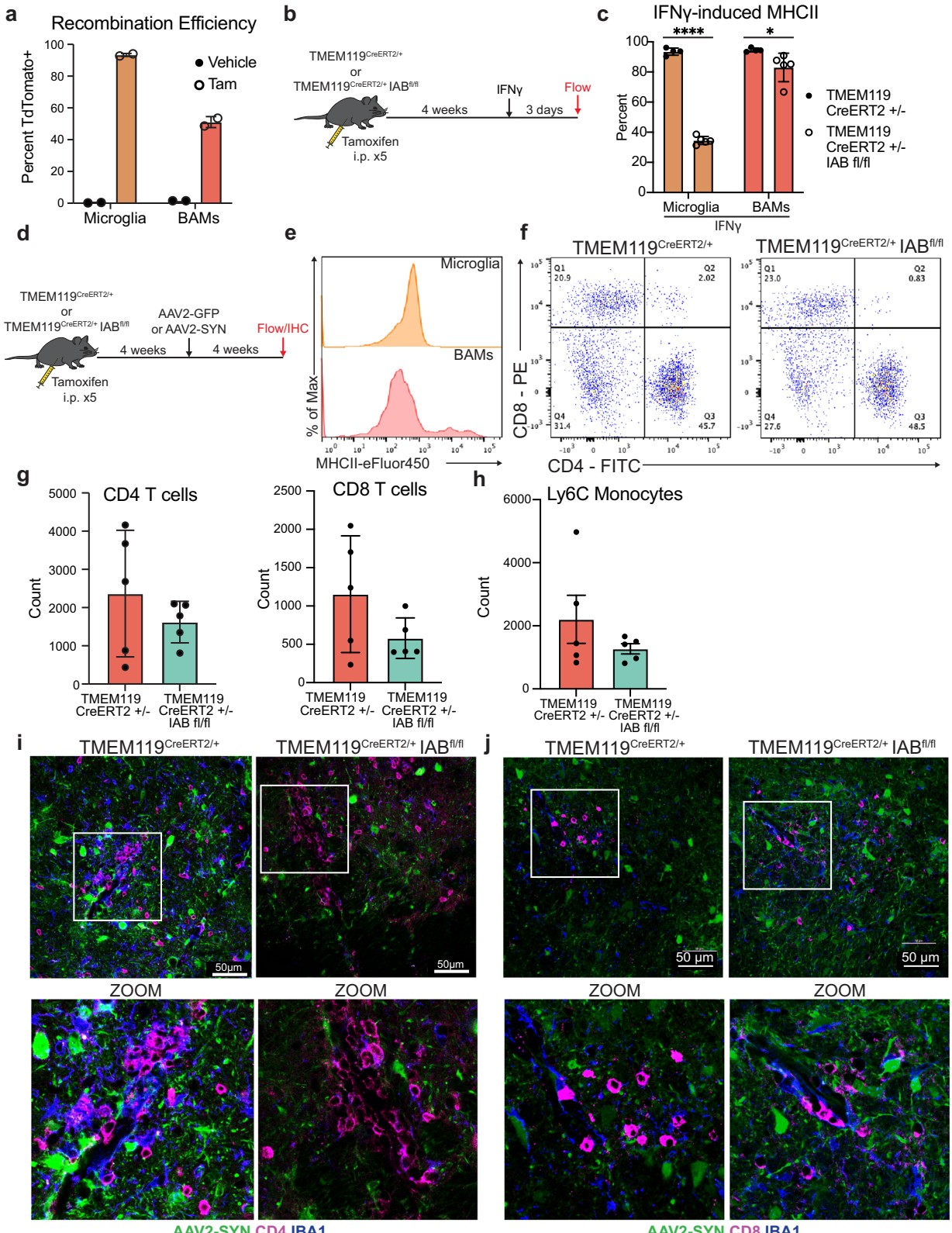

AAV2-SYN  CD4  IBA1                                    AAV2-SYN  CD8  IBA1

importance of combining single-cell sequencing with other approaches, whether genetic or pharmacological, to dissect the functions of unique subsets of CRMs. By focusing exclusively on the CX3CR1⁺ CRMs, we were able to derive sufficient power to resolve the unique activation states of BAMs and identify commonalities and key differences they have with microglia. While the relative activation states of DAMs and DaBAMs seem to have extensive overlap in their

transcriptional response, the unique genes specific to the activation response of each population are likely related to their specific function and cellular localization in the brain. As our ability to more accurately define the immune cell heterogeneity in the CNS increases, future research helping to delineate the specific differences and functions of these unique CRM subsets will be essential in identifying viable routes for targeted therapeutics in CNS disease.

**Fig. 3 | Microglial class II antigen presentation is dispensable for alpha-synuclein-induced T-cell infiltration. a** Fate mapping experiment to determine the baseline and inducible recombination efficiency of CRMs in TMEM119[CreERT2/+] TdTomato[fl/fl]. Percent of cells that are TdTomato+ are displayed. $N = 2$ mice per treatment group. Mean ± SEM is displayed. Representative results of two independent experiments are shown. **b** Experimental paradigm designed to test microglial-specific tamoxifen-induced deletion of the *Iab* locus. Because baseline microglial MHCII expression is low, mice were given intra-nigral IFNγ at 4 weeks post-tamoxifen treatment, and microglial MHCII expression was assayed 3 days post injection. **c** Quantification of MHCII expression by CRMs, including microglia and BAMs, following IFNγ treatment. Percent of cells expressing MHCII are shown. $N = 4$ TMEM119 CreERT2/+ and 5 per TMEM119 CreERT2/+ Iab fl/fl mice per group. Two-way ANOVA with Tukey's multiple comparison test, *$p = 0.0291$, ****$p < 0.0001$, mean ± SD is shown. Representative results of one independent experiment are shown. **d** Experimental paradigm for conditional Iab deletion from TMEM119+ cells. Mice received tamoxifen and 4 weeks later were given AAV2-SYN. Four weeks post AAV2-SYN, flow cytometry and immunohistochemistry were performed. **e** Representative histograms displaying MHCII expression on microglia and BAMs in AAV2-GFP and AAV2-SYN conditions. Y axis represents percent of maximum to allow comparison between differently sized populations. X axis displays MHCII intensity. **f** Representative flow plots of brain-infiltrating CD4+ or CD8+ T cells in TMEM119[CreERT2/+] IAB[fl/fl] mice or TMEM119[CreERT2/+] control mice. Representative results from one independent experiment. **g** Quantification of (**f**). $n = 5$ per group, with one ventral midbrain per n. Mean ± SD is shown. **h** Quantification of infiltrating Ly6C hi monocytes in TMEM119[CreERT2/+] IAB[fl/fl] mice or TMEM119[CreERT2/+] control mice 4 weeks after AAV2-SYN. $N = 5$ per group, with one ventral midbrain per n. Mean ± SD is shown. Representative results from one independent experiment. **i** Immunofluorescent images of CD4+ T cell infiltration in TMEM119[CreERT2/+] IAB[fl/fl] mice or TMEM119[CreERT2/+] control mice. Tissues are labeled with CD4 (magenta), AAV2-SYN (green), and IBA1 (blue). Images taken at ×40 magnification (top) and digitally zoomed (bottom). Representative images from one experiment are shown and meant to corroborate flow cytometry results. **j** Immunofluorescent images of CD8+ infiltration in TMEM119[CreERT2/+] IAB[fl/fl] mice or TMEM119[CreERT2/+] control mice. Tissues are labeled with CD8 (magenta), AAV2-SYN (green), and IBA1 (blue). Images taken at ×40 magnification (top) and digitally zoomed (bottom). Representative images from one experiment are shown and meant to corroborate flow cytometry results. *Source data is provided in the "Source data" file.

While microglia represent the majority of CRMs, our data indicate that their functional role as antigen-presenting cells in the context of an ongoing immune response is limited. Instead, based on their genetic profile and the finding that microglial MHCII is dispensable for α-syn-mediated neuroinflammation, the role of intraparenchymal microglia primarily seems to be for the clearance of cellular debris via phagocytosis, facilitating regeneration, and sensors of tissue damage via pattern recognition receptors such as TLRs. Our findings demonstrate that BAMs rather than microglia play a crucial role in the localization of the immune response and are responsible for the restimulation of CD4+ T cells that is necessary for tissue infiltration and cytokine production. Recent evidence has highlighted the potential importance of classical dendritic cells in the meninges for CNS autoimmune inflammation, whereas BAM depletion in this context provided only partial protection[30]. Surprisingly, our study suggests an alternative process may occur during neurodegenerative inflammation, as neither our genetic nor pharmacological manipulations affected populations of classical dendritic cells.

In previous studies and the studies within, we have focused our investigation on the role of class II-mediated neuroinflammation and neurodegeneration in a preclinical model of PD. In these studies, we found that AAV2-SYN transduced mice deficient in microglial MHCII expression still displayed α-syn pathology in the SN but did not display overt neurodegeneration. Accordingly, our previous studies have also indicated that interventional strategies altering MHCII expression or T-cell infiltration attenuated α-syn-mediated neuroinflammation and neurodegeneration[10,23]. Collectively, these results suggest that neuroinflammation may hasten the neurodegenerative process, potentially exacerbating progression of neuronal dysfunction and TH neuron death. One limitation of our study is the technical capacity for long-term or sustained depletion of BAMs. However, the finding that BAM depletion prevents infiltration of peripheral immune cells, a process that has been shown to drive neurodegeneration in this model, indicates that long-term inhibition of the function of BAMs is likely to attenuate neurodegeneration[10,31]. These findings emphasize the importance of future research on the unique roles these subsets of macrophages may play in both the innate immune response to aggregated proteins as well as any potential disease specific processes including neuronal dysfunction and neurodegeneration.

Numerous pieces of evidence have indicated a potential role for CD3+ T cells in PD based upon the extensive T cell infiltrate seen in post-mortem tissue and strong genetic association between disease risk and HLA status[5,32]. While our understanding of the importance of the adaptive immune system in PD continues to evolve, an important open question remains on the relative contribution of the two unique CD3+ populations. Early studies in the brains of PD patients identified a substantial population of CD8+ T cells, however, recent work in both humans and animal models have found circulating α-syn-responsive CD4+ T cells and indicate that the loss of CD4+ T cells is neuroprotective[10]. These data fit with the established importance of MHCII for antigen presentation, specifically to CD4+ T cells, and its known genetic association with PD. Furthermore, our study found limited changes in infiltrating CD8+ T cells, further emphasizing the contribution of class II-CD4+ T cell interactions in neurodegeneration. Our assessment of postmortem brain from PD patients identified an enhanced presence of CD8+ T cells in the perivascular space, a finding that is not unique to PD. Numerous post-mortem studies of AD and ALS have similarly noted infiltration of cytotoxic T cells[33,34]. As CD8+ T cells have been implicated in tissue remodeling and long-term immune surveillance, these results may reflect the specific stage of disease captured in postmortem tissue. Recent work in humans found that α-syn reactive CD4+ T cells were present in the very early stages of disease and decreased with disease progression[35]. CD8+ T cells have also been implicated in the neuroinflammatory response to α-syn, although their deleterious effects appear to be partially dependent on CD4+ T cell function and independent from BAMs[10,36,37]. Future work attempting to delineate the unique functions of both populations of CD3+ T cells and their association with different disease stages in PD will be essential to modulate and affect the clinical course of disease.

In summary, our study indicates that BAMs, not microglia, specifically mediate neuroinflammation in response to α-syn, contrary to previous research that has focused only on microglia or meningeal dendritic cells. We show that α-syn expression led to an expansion in BAM numbers and to a unique activation state coined here as DaBAMs. These BAMs play an essential role in immune cell recruitment, infiltration, and antigen presentation, a key process prior to α-syn-mediated neurodegeneration. In addition, we have demonstrated increased interactions between BAMs and CD3+ T cells in the perivascular space of human PD brains, indicating that this may be an ongoing process of disease pathogenesis. Future studies will unravel the mechanistic features of this process and understanding these immune processes will be crucial to developing effective disease-modifying treatments for PD.

## Methods

All research described here complies with all relevant ethical regulations and all animal research protocols have been approved by the Institutional Animal Care and Use Committee at the University of Alabama at Birmingham.

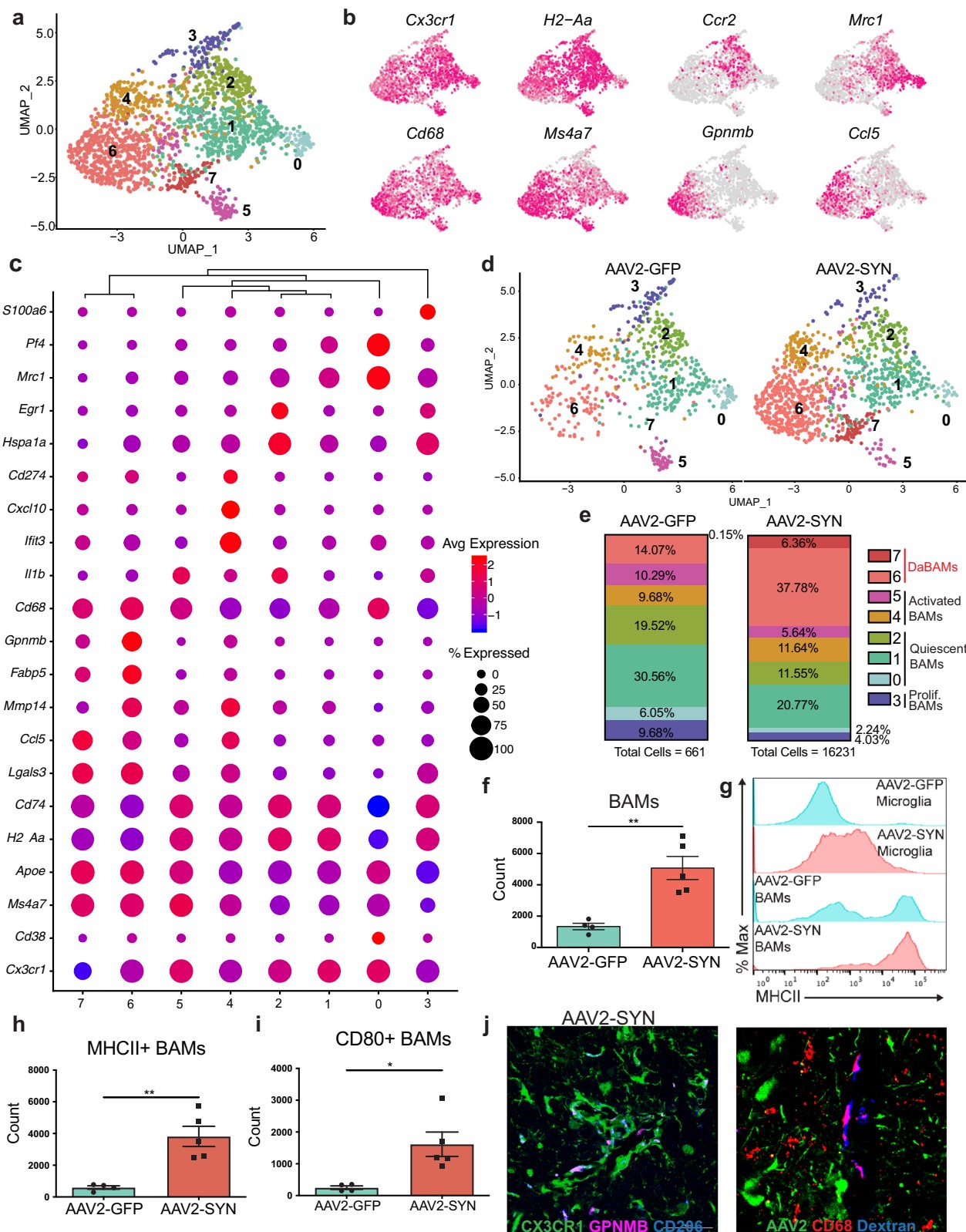

## Mice

Male and female C57BL/6J (#000664 Jackson Laboratories), B6.129P2(Cg)-*Cx3cr1*$^{tm1Litt}$/J, referred to as a CX3CR1 reporter knock-in (#005582 Jackson Laboratories), B6.129P2(C)-*Cx3cr1*$^{tm2.1(cre/ERT2)Jung}$/J, referred to as CX3CR1 CreERT2 (#020940 Jackson Laboratories), C57BL/6-*Tmem119*$^{em1(cre/ERT2)Gfng}$/J, referred to as TMEM119 CreERT2 (#031820 Jackson Laboratories), B6.Cg-*Gt(ROSA)26Sor*$^{tm9(CAG-tdTomato)Hze}$/J,

referred to as TdTomato (#007909, Jackson Laboratories), and B6.129X1-*H2-Ab1*$^{tm1Koni}$/J, referred to as Iab$^{fl/fl}$ (#013181 Jackson Laboratories) were used for these studies. Mice were bred and maintained on a congenic background. Littermate controls were used in most experiments. Sex was not tracked as a biological variable, but distribution of sex was equal across all groups. In instances of pooled ventral midbrains, males and female mice were pooled into one sample

**Fig. 4 | Alpha-synuclein specific changes in border-associated macrophages.**
**a** Cluster 8 from the larger dataset was isolated and subjected to unbiased clustering. Displayed are integrated BAM U-MAP plots colored for expression of genes specifically upregulated in key BAM clusters. **b** Integrated U-MAP plots colored for expression of BAM identity genes and cluster defining genes. **c** Dot plot corresponding to the integrated U-MAP plot of BAMs demonstrating cluster-identifying genes. Dot size represents the percentage cluster cells expressing the gene and dot color represents its average expression within the cluster. **d** U-MAP projections demonstrating BAM cluster changes with AAV2-SYN compared to AAV2-GFP. Overall, eight distinct BAM clusters were identified. **e** Proportion of the total cells in each BAM cluster in response to AAV2-GFP or AAV2-SYN. Cells are labeled according to their genetic profiles. **f** Quantification of flow cytometric data. BAMs were gated as live, CD45$^+$ CD11b$^+$ CX3CR1$^+$ Ly6C$^-$ and CD38$^+$. Unpaired T-test, two-tailed, $n = 4$ AAV2-GFP and 5 AAV2-SYN transduced samples per group, with 2 pooled ventral midbrains per sample. $**p = 0.0033$. Mean ± SD is shown. Representative results from three independent experiments. **g** Representative histograms (left) displaying MHCII expression on microglia and BAMs in AAV2-GFP and AAV2-SYN conditions. Y axis represents percent of maximum to allow comparison between differently sized populations. X axis displays MHCII intensity. **h** Flow cytometric quantification of MHCII on BAMs. BAMs are gated as live, CD45$^+$ CD11b$^+$ CX3CR1$^+$ Ly6C$^-$ and CD38$^+$. Unpaired T-test, two-tailed, $n = 4$ AAV2-GFP and 5 AAV2-SYN transduced samples per group, with 2 pooled ventral midbrains per sample. $**p = 0.003$. Mean ± SD is shown. Representative results from three independent experiments. **i** Flow cytometric quantification of CD80 on BAMs. BAMs are gated as live, CD45$^+$ CD11b$^+$ CX3CR1$^+$ Ly6C$^-$ and CD38$^+$. Unpaired T-test, two-tailed, $n = 4$ AAV2-GFP and 5 AAV2-SYN transduced samples per group, with 2 pooled ventral midbrains per sample. $*p = 0.0167$. Mean ± SD is shown. Representative results from two independent experiments are shown. **j** Immunofluorescence confirming expression of GPNMB and CD68 in perivascular BAMs. Tissue is labeled with CX3CR1 (green), GPNMB (left) or CD68 (right) (magenta), and Cd206 (blue). Images are taken at ×60 magnification. Representative images are shown from one experiment and corroborate scRNA sequencing and flow cytometry findings. *Source data is provided in the "Source data" file.

to attain adequate cell numbers. Mice were housed in clean filter-top cages and kept on a 12 h light/dark cycle with an ambient temperature of 25 °C. They received food and water ad libitum. All animal research protocols were approved by the Institutional Animal Care and Use Committee at the University of Alabama at Birmingham.

## AAV2-SYN and AAV2-GFP vector
α-syn or green fluorescence protein (GFP) overexpression was mediated via adeno-associated virus (AAV) transduction of neurons in the SNpc. Construction and purification of the rAAV vectors, rAAV-CBA-IRES-EGFP-WPRE (CIGW, AAV2-GFP) and rAAV-CBA-SYNUCLEIN-IRES-EGFP-WPRE (CISGW, AAV2-α-SYN) are described in detail previously[22]. For experiments within, AAV2-SYN and AAV2-GFP were manufactured and purified by the University of Iowa Viral Vector Core. Both AAV2-GFP and AAV2-SYN were stereotaxically injected into the SNpc of mice at a titer of $2.6 × 10^{12}$ vg/mL[22,38]. Viral titer was determined by PCR.

## Stereotaxic surgery
Surgical procedures were performed as previously described[11]. Briefly, mice were anesthetized with inhaled isoflurane and immobilized in a stereotaxic frame. A volume of 2 μL AAV2-SYN or AAV2-GFP was injected into the substantia nigra pars compacta at 0.5 μL per minute with a Hamilton syringe. The needle was left in the brain after injection for an additional 3 min to allow diffusion, and then slowly retracted. Mice for immunohistochemistry and unbiased stereology received unilateral injections, to provide an internal control for surgeries. Mice designated for flow cytometry received bilateral injections. The stereotaxic coordinates used from bregma were: AP −3.2 mm, ML ± 1.2 mm, and DV −4.6 mm from dura.

Mice included in the liposome-depletion studied received bilateral injections into the lateral ventricles of 10 μL saline or clodronate-filled liposomes (20 μL total volume per animal at a concentration of 5 mg/mL) at a rate of 1 μL per minute with a Hamilton syringe. The needle remained in place for an additional 4 minutes and then was slowly retracted. The stereotaxic coordinates for the lateral ventricles used from bregma were: AP + 0.3 mm, ML ± 1.0 mm, and DV −2.7 mm from dura. Liposomes were obtained from Encapsula Nanoscience, LLC (Standard Macrophage Depletion Kit, cat. # CLD89012ML).

## Tamoxifen treatment
Mice in fate mapping experiments (CX3CR1$^{CreERT2/+}$ TdTomato) and conditional knock-out experiments (CX3CR1$^{CreERT2/+}$ Iab$^{fl/fl}$) received two doses of tamoxifen, injected subcutaneously 48 h apart, as previously described[14,16]. Doses were 4 mg tamoxifen dissolved in 200 L of corn oil. Recombination efficiency was determined using fate mapping mice. Mice rested 6 weeks after tamoxifen treatment before AAV injection to allow for full monocyte and DC turnover. At this time

point, recombination efficiency in microglia (90.25%), BAMs (83.68%), and cDCs (13.4%) was determined.

For microglial-specific manipulation, Tmem119$^{CreERT2/+}$ Iab$^{fl/fl}$ mice were given five doses of tamoxifen, injected intraperitoneally daily. Doses were 2 mg dissolved in 100 μL of corn oil. Mice rested 4 weeks after tamoxifen treatment before AAV injection to allow for clearance of tamoxifen. At this time point, fate mapping mice were used to determine the recombination efficiency for microglia (~93.4%) and BAMs (~45.7%). IFNγ injection was used to more accurately determine ability to express MHCII, as microglia do not express high levels at baseline. With IFNγ, MHCII expression was depleted in microglia (34.5% of Tmem119$^{CreERT2/+}$ Iab$^{fl/fl}$ mice could express MHCII compared to 93% of controls) but largely remained in BAMs (83% of Tmem119$^{CreERT2/+}$ Iab$^{fl/fl}$ mice could express MHCII compared to 94.6% of controls)

## Immunohistochemistry of mouse samples
Four weeks or six months post-AAV injection, mice were deeply anesthetized, euthanized, and brains and meninges were collected for processing. In short, animals were transcardially perfused with 0.01 M PBS containing heparin, followed by 4% paraformaldehyde. Brains were extracted, drop-fixed overnight at 4 °C, and transferred to 30% sucrose in 0.01 M PBS for cryoprotection for 48 h. Brains were frozen on dry ice and sectioned using a sliding microtome. Coronal sections 40-μm thick were serially collected and stored in 50% glycerol in 50% 0.01 M PBS at −20 °C.

For analysis using fluorescence, sections were washed with Tris-buffered saline (TBS) and blocked in 5% normal serum for 1 h. Following, sections were labeled with anti-MHCII (clone M5/114.15.2, eBiosciences, cat #50-112-9455, used at 1:500 dilution), anti-α-synuclein (phospho-Serine129, clone EP1536Y, Abcam, cat #ab51253, used at 1:2000 dilution), anti-tyrosine hydroxylase (TH, Sigma-Aldrich, cat # AB152, used at 1:500 dilution), anti-IBA1 (polyclonal, Wako, cat # NC1718288, used at 1:500 dilution), anti-CD4 (clone RM4-5, BD Bioscience, cat # BDB553043, used at 1:500 dilution), anti-CD8 (clone 4SM15, eBioscience, cat # 14-0808-82, used at 1:500 dilution), anti-CD206 (clone C068C2, Biolegend, cat #141701, used at 1:100 dilution), anti-GPNMB (R&D Systems, cat. #AF2330, used at 1:100 dilution), anti-APOE (Sigma-Aldrich, cat. #AB947, used at 1:100 dilution), anti-CD31 (clone 2H8, Invitrogen, cat #ENMA3105, used at 1:500 dilution), and anti-laminin (Sigma-Aldrich, cat #L9393-100UL, used at 1:1000 dilution). Antibodies were diluted in 1% normal serum in TBS-Triton (TBST) overnight at 4 °C. Sections were washed and incubated with the appropriate Alexa-fluor conjugated secondary antibodies (Life Technologies and Jackson Immunoresearch) diluted in 1% normal serum and TBST for 2 h. The following fluorescent conjugated secondaries were used: Alexa Fluor 594 donkey anti-rat (Invitrogen, cat #A21209,

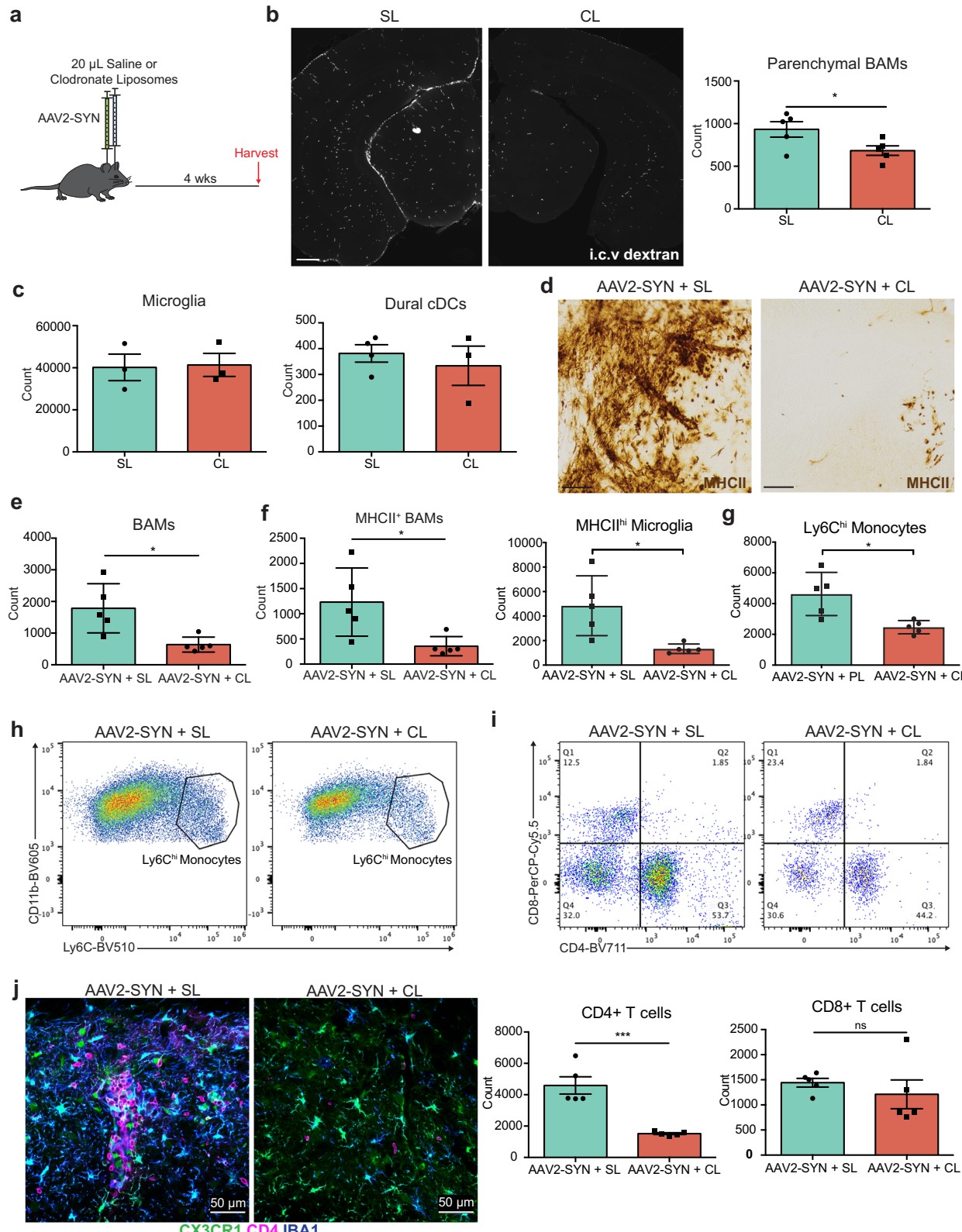

used at 1:1000 dilution), Alexa Fluor 647 donkey anti-rabbit (Invitrogen, cat #A31573, used at 1:1000 dilution), Alexa Fluor 647 donkey anti-rat (Jackson Immunoresearch, cat #712-605-153, used at 1:500 dilution), Alexa Fluor 555 donkey anti-rabbit (Invitrogen, cat # A31572, used at 1:1000 dilution). Sections were mounted onto coated glass slides, and coverslipped using hard set mounting medium to preserve fluorescent signal (Vector Laboratories).

For analysis using diaminobenzidine (DAB) staining, sections were washed in TBS, quenched in 0.03% hydrogen peroxide, and blocked in 5% normal serum for 1 h. Following this, sections were labeled with anti-MHCII (clone M5/114.15.2, eBiosciences, cat #50-112-9455, used at 1:500 dilution), anti-α-synuclein (phospho-Serine129 clone EP1536Y, Abcam, cat #ab51253, used at 1:2000 dilution), and anti-CD4 (clone RM4-5, BD Bioscience, cat # BDB553043, used at 1:500 dilution).

**Fig. 5 | Specific depletion of border-associated macrophages prevents alpha-synuclein induced neuroinflammation. a** Experimental paradigm for Clodronate Liposome (CL) BAM depletion in combination with AAV2-SYN administration. Flow cytometry and IHC are performed 4 weeks post administration. **b** Left: Tiled fluorescent images demonstrating BAM depletion within the ventral midbrain 7 days after CL or Saline Liposome (SL) administration. BAMs are marked with dextran that was administered i.c.v. 24 h prior to sacrifice (white). Right: flow cytometric quantification of BAM depletion with CL. BAMs are gated as live, CD45$^+$ CD11b$^+$ CX3CR1$^+$ Ly6C$^-$ and CD38$^+$. Scale bar is 500 μm. Unpaired T-test, two-tailed, $n = 5$ samples per group with 2 ventral midbrains pooled per sample. *$p = 0.0466$. Mean ± SD is shown. Representative results from two independent experiments. **c** Flow cytometric quantification 7 days post CL or SL administration demonstrating that parenchymal microglia ($N = 3$ samples per group) and dural cDCs ($N = 4$ SL and 3 CL samples per group) are unaffected with treatment. Each sample comprised of 2 pooled ventral midbrains. Mean ± SD is shown. Representative results from two independent experiments. **d** Immunohistochemistry depicting MHCII (brown) expression within the ventral midbrain after AAV2-SYN + SL or AAV2-SYN + CL, demonstrating that BAM depletion prevents MHCII expression in the midbrain. Displayed images are cropped from larger tiled ones. Scale bar is 50 μm.

Representative images are displayed from two independent experiments. **e** Confirmation of BAM depletion in AAV2-SYN with CL via flow cytometry. Unpaired T-test, two-tailed, $n = 5$ per group. *$p = 0.0134$. Mean ± SD is shown. Representative results from two independent experiments. **f** Flow cytometric quantification of MHCII$^+$ BAM and MHCII$^+$ microglia reduction with CL treatment. Unpaired T-test, two-tailed, $n = 5$ per group. *$p = 0.0238$ and 0.013, respectively. Mean ± SD is shown. Representative results from two independent experiments. **g** Quantification of infiltrating Ly6C$^{hi}$ monocytes in response to α-syn. Monocytes are gated as live, CD45$^+$ CD11b$^+$ CX3CR1$^-$ CD38$^-$ Ly6C$^{hi}$. Unpaired T-test, two-tailed, $n = 5$ per group. *$p = 0.011$. Mean ± SD is shown. Representative results from two independent experiments. **h** Representative flow plots of (**g**). **i** Top: Representative flow plots of CD4$^+$ and CD8$^+$ T cell infiltration with SL or CL treatment. Below: Quantification of flow cytometry demonstrating that BAM depletion reduces CD4$^+$ T cell infiltration. Unpaired T-test, two-tailed, $n = 5$ per group. ***$p = 0.0005$. Mean ± SD is shown. Representative results from two independent experiments. **j** Immunofluorescent images demonstrating reduced CD4$^+$ T cell infiltration and myeloid activation with CL-mediated BAM depletion. Tissues are labeled with CX3CR1 (green), CD4 (magenta), and IBA1 (blue). Images were captured at ×40. *Source data is provided in the "Source data" file.

Antibodies were diluted in 1% normal serum in TBST and sections were incubated overnight at 4 °C. Appropriate biotinylated secondary antibodies (Jackson ImmunoResearch, used at 1:500 dilution) were diluted in 1% normal serum in TBST and incubated for 2 h at room temperature. R.T.U. Vectastain ABC reagent (Vector Laboratories) was applied, followed by the DAB kit (Vector Laboratories) according to manufacturer's instructions. Sections were mounted on coated glass slides, dehydrated in increasing concentrations of ethanol, cleared using Citrisolv, and coverslipped with Permount mounting medium (Fisher).

### Confocal imaging
Confocal images were acquired using either a Leica TCS-SP5 laser scanning confocal microscope, or a Nikon Ti2-C2 confocal microscope. Images were exported and processed using Adobe Photoshop and Illustrator.

### Brightfield and fluorescent imaging and quantification in mice
Tiled brightfield and fluorescent images were acquired using a Nikon Ni-E microscope. All sections within an experiment were captured using the same settings. Three sections per brain encompassing the SNpc were imaged and representative images were selected. Brightfield images were exported and utilized without processing. Fluorescent images were exported and processed using Adobe photoshop and Illustrator. $n = 3–6$ animals were examined per treatment group. For T cell quantification, tiled brightfield images of ventral midbrains were acquired. ImageJ software was used to draw a region of interest containing the injected ventral midbrain, and T cells within the ROI were counted. Number of T cells was divided by the area of the ROI to obtain cells/mm$^2$.

### Unbiased stereology
For quantification of TH neurons in the SNpc, unbiased stereology was performed as previously described[11,23]. TH-DAB stained SNpc slides were coded and analyzed by a reviewer blinded to condition. An Olympus BX51 microscope with MicroBrightfield software was utilized. A total of four to five sections encompassing the rostrocaudal SNpc were quantified using the optical fractionator method using the StereoInvestigator software (v2021.1.3). Both ipsilateral injected and contralateral uninjected sides of the SNpc were quantified. TH$^+$ neurons within the contours of the SNpc on a 100 μm × 100 μm grid were counted using an optical dissector height of 22 μm. Weighted section thickness was used to account for variations in tissue thickness. Brightfield images of TH neurons in the SNpc were acquired using the Nikon Ni-E microscope.

### Mononuclear cell isolation
Four weeks after bilateral transduction of AAV2-SYN or AAV2-GFP, mononuclear cells within the ventral midbrain of mice were isolated as previously published[10,11]. A 3 mm section of the midbrain was isolated, manually dissociated, and digested with Collagenase IV (1 mg/mL, Sigma) and DNAse I (20 μg/mL, Sigma) diluted in RPMI 1640 (Sigma). Digested tissue was passed through a 70-μm filter to obtain a single-cell suspension, and mononuclear cells were isolated using a 30/70% Percoll gradient (GE). The resulting interphase layer was collected for analysis.

Isolated mononuclear cells were blocked with anti-Fcγ receptor (clone 2.4G2 BD Biosciences, cat #BDB553141, used at 1:100 dilution) and surface stained with fluorescently conjugated antibodies. Multiple fluorophores were used throughout the paper, but antibody clones and vendor remained consistent regardless of fluorophore. Antibodies against CD45 (Clone 30-F11, eBioscience cat #50-112-9701, used at 1:500 dilution), CD11b (Clone M1/70, BioLegend cat #101237, used at 1:250 dilution), CX3CR1 (Clone SA011F11, BioLegend cat #149008, used at 1:500 dilution), Ly6C (clone HK 1.4, BioLegend cat #128015, used at 1:500 dilution), CD38 (clone 90, Biolegend cat #102717, used at 1:500 dilution), MHCII (clone M5/114.15.2, BioLegend cat #107607, used at 1:500 dilution), CD80 (Clone 16-10A1, BD Horizon cat #BDB612773, used at 1:250 dilution), PD-L1 (clone B7-H1, BioLegend cat #124319, used at 1:250 dilution), CD4 (clone GK1.5, BioLegend cat #100447, used at 1:250 dilution), and CD8a (clone 53-6.7, BioLegend cat #100734, used at 1:250 dilution) were used. A fixable viability dye was also used according to manufacturer's instructions (Fixable Near-IR LIVE/DEAD Stain Kit, Life Technologies, cat #NC0584313 or Fixable Blue Dead Cell Stain Kit for UV excitation, Invitrogen, cat #50-112-1524, used at 1:500 dilution).

For staining of T cell intracellular cytokines, cells were first stimulated with phorbol myristate acetate (PMA, 50 ng/mL, Fisher BioReagents, cat. #50-058-20001) and ionomycin (750 ng/mL, Millipore Sigma, cat #AAJ62448MCR) in the presence of GolgiStop (1:1000, BD Biosciences, cat #BDB554715) for 4 h at 37 °C with 5% $CO_2$. Cells were then blocked, surface stained, and processed using the BD Cytofix/Cytoperm Staining Kit (BD Biosciences) according to instruction manuals, and cells were stained with the fluorescently conjugated antibodies against IFNγ (clone XMG1.2, eBioscience cat # 50-111-85, used at 1:100 dilution), IL-4 (clone 11B11, BioLegend cat #504117, used at 1:100), IL-17a (clone eBiol7B7, eBioscience cat #50-112-9052, used at 1:100), and IL-10 (clone JES5-16E3, BioLegend cat #505005, used at 1:100). For staining of intracellular myeloid markers, cells were processed using the BD Cytofix/Cytoperm Staining Kit according to instruction manuals and stained with CD68 (clone FA-11, Biolegend cat #137024, used at 1:500 dilution) and Ki67 (clone 16A8, Biolegend cat

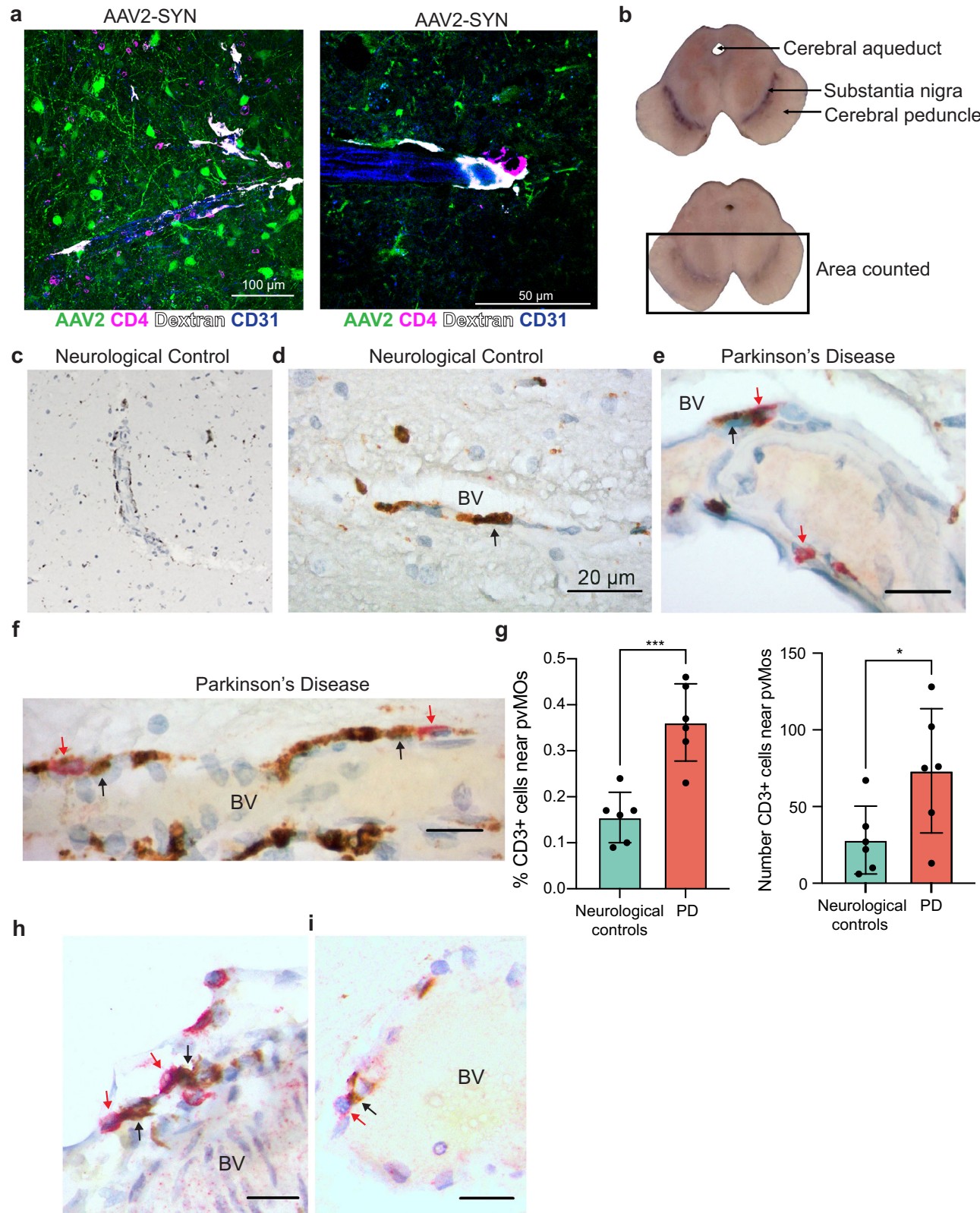

#652413, used at 1:250 dilution). Samples were run on an Attune Nxt flow cytometer (Thermo Fisher Scientific) or a BD Symphony (BD Biosciences) and analyzed using FlowJo software (Tree Star).

**Single-cell RNA sequencing**
Mononuclear cells were isolated for sequencing as described above with 3–4 ventral midbrains pooled per sample, and sorted for CD45$^+$

CD11b$^+$ CX3CR1$^+$ Ly6C$^-$ Ly6G$^-$ and NK1.1$^-$ cells on a BD FACsAria. Sorted cells were loaded onto the 10X Chromium platform (10X genomics), and libraries were constructed using the Single Cell 3′ Reagent Kit V2 according to the manufacturer's instructions. Two biological replicates for each group were processed separately. Samples were sequenced to an average depth of 20,000 reads per cell on an Illumina NovaSeq. Sequencing files were processed, mapped to mm10, and count

**Fig. 6 | Border-associated macrophages in human PD. a** Immunofluorescence showing abundance of close interactions between CD4$^+$ T cells and perivascular macrophages in the mouse SNpc. Tissue is labeled with AAV2-SYN (green), CD4 (magenta), Dextran (white), and CD31 (blue). Images are taken at ×40 (left) and ×60 (right). **b** Transverse sections of midbrains from one neurological control (top) and one patient with PD (bottom), showing the anatomic landmarks of the cerebral aqueduct, substantia nigra, and cerebral peduncle. Rectangle denotes the region of interest where CD3/CD68 double immunostaining was counted. **c** Images of CD68$^+$ BAMs located in the perivascular space of the substantia nigra of non-neurological disease controls, demonstrating their abundance in human brains. **d** High magnification demonstrating elongated, vessel-localized CD68$^+$ BAMs in the non-neurological disease control substantia nigra. Black arrows point to BAMs. BV denotes the blood vessel. Scale bar is 20 μm. **e** Vessel-localized CD68$^+$ BAMs (brown) and CD3$^+$ T cells (red) in the substantia nigra of human PD (right). Black arrows denote BAMs, red arrows denote CD3$^+$ T cells. BV indicates the blood vessel. Scale bars are 10 μm. **f** Vessel-localized CD68$^+$ BAMs (brown) and CD3$^+$ T cells (red) in the substantia nigra of human PD (right). Black arrows denote BAMs, red arrows denote CD3$^+$ T cells. BV indicates the blood vessel. Scale bars are 10 μm. **g** Quantification of BAM − T cell interactions in human PD. The percentage of perivascular CD3$^+$ cells adjacent to perivascular CD68$^+$ cells was divided by the total number of CD3$^+$ cells in the ventral midbrain of 6 DLBD patients and 6 controls. Graph displays mean values ± SD. Unpaired T test, two-sided, $*p = 0.0375$, $***p = 0.0005$. Mean ± SD is shown. **h** Vessel-localized CD8$^+$ T cells (red) and CD68$^+$ BAMs (brown) cells in the substantia nigra of human PD. Black arrows denote BAMs, whereas red arrows denote CD8$^+$ T cells that are in close proximity to BAMs. BV denotes blood vessel. Scale bars are 10 μm. Representative images of one experiment are displayed. **i** Vessel-localized CD4$^+$ T cells (red) and CD68$^+$ BAMs (brown) cells in the substantia nigra of human PD. Black arrows denote BAMs, red arrows denote CD4$^+$ T cells BV denotes blood vessel. Scale bars are 10 μm. Representative images of one experiment are displayed. *Source data is provided in the "Source data" file.

matrices were extracted using the Cell Ranger Single Cell Software (v 3.1.0).

## Single-cell RNaseq analysis

Analyses were performed in R using Seurat (v3.1). Data was pre-processed by removing genes expressed in fewer than 3 cells and excluding cells that were outliers for the number of RNA molecules or more than 12% mitochondrial genes. The datasets were merged together and integrated following the Seurat standard integration method using the 2000 most variable genes. Following normalization, UMAP dimensional reduction was performed using the first 20 principal components. Clustering was performed following identification of nearest neighbors, using the first 20 dimensions and a resolution of 0.7. Clusters predominantly composed of either cells with low RNA content, doublet cells, or non-macrophage lineage cells were removed and the datasets were re-clustered following dimensional reduction. Marker genes for each cluster were determined using the FindAllMarkers function of Seurat with a minimum Log2 fold change threshold of +/−0.25 with the Wilcoxon ranked-sum test. For the BAM re-analysis, BAMs were identified in the merged group files and then integrated as described above. Clustering was performed as described above with a resolution of 0.7 using the first 20 dimensions.

## Pseudotime analysis

Filtered and merged datasets were imported into Monocle3 by generating a cell dataset from the raw counts slot of the Seurat object on the 2000 most variable genes. Normalization and PCA were done with the preprocess_cds command from Monocle3 using the first 50 dimensions, and batch correction was applied using the align_cds command, which utilizes the Batchelor tool (Haghverdi et al.[39]). UMAP dimensional reduction was performed using the reduce_dimension command. Cells were clustered with cluster_cells using "Louvain" with the k set to 20. The trajectory graph was learned on the Monocle-derived clusters by calling learn_graph. Cells on the UMAP plot are colored by Seurat-derived clusters. Pseudotime was determined using the quiescent clusters as the starting point.

## Pathway analyses

Upregulated genes in clusters of interest were uploaded to the WEB-based GEne SeT AnaLysis Toolkit (WebGestalt) database. Enrichment analyses, including GO and KEGG analyses, were performed using Hypergeometric testing and a Benjamini Hochberg correction for multiple testing. The top 10 pathways with the most significant $p$ values and 2 or more genes in the group were identified and displayed.

## Analysis of human postmortem tissues

Consent for autopsy was obtained from patient legal surrogates through standardized consenting procedures. Human postmortem brain tissue was obtained from the New York Brain Bank. Protocols were approved by the Institutional Review Board of Columbia University Medical Center. After brains were fixed in formalin for 10–14 days, the brainstems were removed, cut in a transverse plane, and sections embedded in paraffin blocks. Blocks of midbrains were cut at a thickness of 7 μm and initially stained with H&E and immunostained for α-synuclein. The diagnosis of DLBD was based upon finding Lewy bodies in brainstem structures including the substantia nigra and also in the cerebral cortex. Control patients had no history of PD and their brains did not contain Lewy bodies. Additional sections from the same blocks were immunostained for CD3 and CD68, using a double immunostaining procedure. CD3 (clone LN10, Leica cat #CD3-565-L-CE, used at 1:100), CD4 (clone 4B12, Leica cat # CD4-368-L-CE, used at 1:100), and CD8 (clone 4B11, Leica cat #CD8-4B11-L-CE, used at 1:100) were stained using a Bond automated stainer, and visualized with alkaline phosphatase reagent and BOND polymer Refine Detection Kit (red); CD68 (PG-M1, DAKO, cat #ab783, used at 1:100) was stained using a Ventana automated stainer, and visualized peroxidase reagent and the DAKO ultraView Universal DAB Detection Kit (brown).

To count cells, the ventral midbrain, containing the substantia nigra, was marked off by a line that ran from the most dorso-lateral edge of the nigra at one side to the edge at the other side. All cell counts were performed within the area ventral to that line. That way, all vessels entering the nigra from the penetrating branches of the basilar artery would be contained in this area. However, the absolute numbers of blood vessels in any section differed from section to section, and thus the numbers of CD3$^+$ cells in the perivascular spaces varied from section to section. Therefore, it did not seem appropriate to compare the absolute numbers of CD3$^+$ cells from section to section and case to case. Instead, we chose to count all CD3$^+$ cells in a ventral section and express the ratio of the numbers of these cells that lay adjacent to CD68$^+$ BAMs to the total number of CD3$^+$ cells in the section. We did not count any CD3$^+$ cells within blood vessels or within the midbrain parenchyma.

## Statistical analysis

Flow cytometry experiments utilized three to five independent samples per group (precise numbers are listed in figure legends), with two ventral midbrains per sample (i.e., each experiment utilized 6–10 mice per group). All statistical analyses were performed using GraphPad Prism 9 software (v9.3.1). Data were analyzed using an unpaired t-test (two-tailed) or two-way ANOVA with Tukey's multiple comparison test. Graphs display the individual values and mean ± SEM, with $*p < 0.05$, $**p < 0.01$, $***p < 0.0005$, $****p < 0.0001$.

## Reporting summary

Further information on research design is available in the Nature Portfolio Reporting Summary linked to this article.

## Data availability

The single-cell RNA sequencing data is available publicly from the NCBI's Gene Expression Omnibus and are accessible at the following link and through the GEO accession number GSE178498. The datasets generated during and/or analyzed during the current study are publicly available in the ASAP data repository on zenodo.org at the following hyperlinks: https://doi.org/10.5281/zenodo.7349228. This dataset contains the quantification of T cell counts immunolabelled in the ventral midbrains of CX3CR1[Cre/+] and CX3CR1[Cre/+] IAB[flox/flox] mice transduced with AAV2-SYN[40]. https://doi.org/10.5281/zenodo.7349256. This dataset contains stereological quantification of TH[+] neurons in the SNpc of CX3CR1[Cre/+] or CX3CR1[Cre/+] IAB[fl/fl] mice transduced with AAV2-SYN or AAV2-GFP[41]. https://doi.org/10.5281/zenodo.7352291. This file contains quantification of interactions between CD3[+] and CD68[+] cells in the perivascular space of postmortem tissue from PD or neurological control midbrains[42]. https://doi.org/10.5281/zenodo.7352402. These files contain raw flow cytometry files from Tmem119[CreERT2/+] or Tmem119[CreERT2/+] IAB[fl/fl] mice at 4 weeks post-transduction with AAV2-SYN[43]. https://doi.org/10.5281/zenodo.7352955. These flow cytometry files were obtained from mice given either clodronate or saline liposomes i.c.v. to test depletion of BAMs in the brain and meninges. A workspace file is included to clarify gating and quantification[43]. https://doi.org/10.5281/zenodo.7443697. This dataset contains flow cytometry files from C57Bl/6 mice transduced with AAV2-SYN or AAV2-GFP at 4 weeks post transduction[44]. https://doi.org/10.5281/zenodo.7352495. These files contain quantification of recombination efficiency in tamoxifen-injected CX3CR1 CreERT2 TdTomato mice using flow cytometry[45]. https://doi.org/10.5281/zenodo.7706957. This dataset contains recombination data of Tmem119 CreERT2/+ TdTomato fl/fl mice injected with tamoxifen or corn oil as a control. https://doi.org/10.5281/zenodo.7709868 [46]. These files contain flow cytometric data demonstrating microglia and BAM ability to upregulate MHCII in Tmem119 CreERT2/+ or Tmem119 CreERT2/+ IAB fl/fl mice injected with intra-striatal IFNγ[47]. https://doi.org/10.5281/zenodo.7714308. This dataset contains .fcs files from AAV2-SYN transduced animals given clodronate or saline liposomes i.c.v. and transduced with AAV2-SYN. This experiment specifically investigated T helper subsets[48]. https://doi.org/10.5281/zenodo.7734867. This dataset contains files from flow cytometry of mice transduced with AAV2-SYN and given i.c.v. liposomes[49]. Source data are provided with this paper.

## Code availability

The code used to analyze single-cell RNA sequencing data is publicly available in the ASAP data repository on zenodo.org with the identifier https://doi.org/10.5281/ZENODO.7693103, (https://doi.org/10.5281/zenodo.7693103)[50].

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

## Acknowledgements
The authors thank W. Wong for the maintenance of the TMEM$^{\text{CreERT2}}$ mouse line, J. Randolph for maintenance of all other mouse lines, and Karen Eskow-Jaunarajs for critical reading of this manuscript. We would also like to thank Dave Sulzer and ASAP "Team Sulzer" for their critical input on data, and the Columbia University Medical Center Department of Pathology and Cell Biology Immuno-stain Lab. Finally, we would like to thank the staff of the New York Brain Bank and Dr. Jean Paul Vonsattel for providing tissues for the human midbrain analysis and the anatomic images of the human midbrain. This work was supported by grants from the Aligning Science Across Parkinson's (grant no. 000375), Michael J. Fox Foundation for Parkinson's Research, the Parkinson Association of Alabama, the National Institute of Neurological Disease and Stroke (grant no. 5 F31 NS106722-02), the National Research Service Award (grant no. 5T32GM109780-02), the National Institute of Neurological Disease and Stroke (grant no. R01NS107316), the Alabama Udall Center (grant no. P50108675), and the Comprehensive Flow Cytometry Core (NIH P30 AR048311 & NIH P30 AI27667).

## Author contributions
A.M.S. and D.A.F. contributed equally. A.M.S., D.A.F., and A.S.H. designed the studies together and wrote the manuscript. A.M.S. and A.J. were responsible for stereotaxic surgery procedures. A.M.S., G.P.W., N.J.G., A.J., J.M.W., and G.M.C. executed IHC and flow cytometry experiments. Flow cytometric and IHC analysis were performed by A.M.S., G.P.W., and A.S.H. A.J. performed unbiased stereology. D.A.F., A.M.S., and A.S.H. designed the scRNA sequencing study, and D.A.F. analyzed and created figures for this data. J.E.G. performed the immunostaining and analysis on human postmortem tissues. A.M.S., D.A.F., and A.S.H. designed and drafted the manuscript figures. D.G.S. and J.E.G. participated in study design and edited the manuscript. All authors read and approved the final manuscript.

## Competing interests
Dr. Standaert has served as a consultant for or received honoraria from Abbvie Inc., Curium Pharma, Appello, Theravance, Sanofi-Aventis, Alnylam Pharmaceutics, Coave Therapeutics, BlueRock Therapeutics, and F. Hoffman-La Roche. All remaining authors declare no competing interests.
