## [Peer Review File · Nature Communications]

Border-associated macrophages mediate the neuroinflammatory response in an alpha-synuclein model of Parkinson diseaseREVIEWER COMMENTS

Reviewer #1 (Remarks to the Author):

In their study Schonhoff and colleagues are trying to define a role of CNS macrophages in the interfaces in a PD mouse model. They used several state-of-the art methods such as scRNA-seq profiling as well as Cx3CR1 and TMEM119 Cre models to deplete MHC class II from tissue resident macrophages. They finally conclude that CNS macrophages in the interfaces rather than microglia shape CD4 interactions by regulating antigen presentation in vivo.

Despite the relevance of the topic there are major weaknesses in the study, especially in the conclusion that MHC class II expression by microglia is redundant for disease (see main comments 1). After improving these points the manuscript will be an important for the field.

Major points

1. Fig. 3 is problematic. Since TEMEM119-mediated depletion of MHC class II is incomplete they authors can't conclude that antigen expression by microglia is not required in this PD model. This might simply be due to the insufficient gene deletion. To circumvent this problem the authors should definitively use either another microglia-specific Cre line (e.g. Hexb ERT2 Cre) or a BAM/CAM line (Mrc1 ERT2 Cre).
2. The transcriptional description of BAMs/CAMs in Fig. 2 (scRNA-seq) is unclear. These cells should be characterized by their core genes such as Mrc-1, CD163, Pf4 etc. This has to be shown.
3. There are no extensive transcriptional differences between BAMs/CAMs. This statement is simply wrong. Very few genes (such as P2yR12, TMEM119, Mrc-1 etc.) are different among the approx. 20.000 genes that a cell has. Please correct this sentence.
4. Figure 6 is especially important of the study because it shows T cell location next to BAMs/CAMs. However, this figure is problematic by two reasons. First, not enough human samples were assessed. At least 8 samples (patients) per condition should be taken. Second, the shown images lack anatomical information of the midbain/sunstantia nigra. Please provide images showing this.

Minor points:

5. Non-microglia macrophages can be called BAMs or CAMs (CNS-associated macrophages). I personally prefer the latter because it clearly indicates the anatomical location. Please give one citation for the CAM literature. Border macrophage are present in the whole body.
6. The sentence on distinct preogenitors of BAM/CAM in the yolk sac is wrong. There is a non-committed cell in the YS that can still give raise to both micorlgia and CAM. Therefore, remove the citation #14 (Utz et al.)
7. Introduction & across the whole manuscript: The term neuroinflammation is used too sloppy. According to a very recent and important consensus paper in microglial nomenclature in Neuron 2022 by Paolicelli and many others (please cite this work), this term should be reserved for conditions of robust influx of circulating cells such as MS, stroke etc. rather than for classical neurodegenerative conditions such as AD, PD, ALS. I truly believe that T cell infiltrates in mouse models of neuroinflammation are highly artificial.
8. Fig. 1 D: white balance is needed for these images. Moreover, the magnification of the images is suboptimal.
9. There is no evidence for different populations of BAMs rather than different clusters representing different states. Please reword these sentences.

Reviewer #2 (Remarks to the Author):

This paper describes for the first time the role of Border Associated Macrophages (BAMs) in the context of a Parkinson disease (PD) model which involves overexpression of the disease-associated (and aggregation prone) protein alpha-synuclein (aS).

The paper describes an extensive and important set of experiments which include aS overexpression, selective depletion of sub populations of immune cells, neuropathology, single cell RNAseq data. Most experiments are in mouse models but the study has incorporated human brain tissue samples. It is well written and the results are well documented. Because the role of BAMs in PD is not clearly defined, there is a significant impact on the field. Overall, studies that demonstrate a role for the immune system in PD are important because they provide fundamental insights into disease mechanisms and the origins of the disease, as well as can highlight novel potential therapeutic targets. Notably, the authors suggest that microglia are not the primary antigen presenting cells in this context, but that BAMs subserve this role. This is a key, potentially paradigm-shifting, finding.

While it can be tempting for a referee to suggest that cytokines were measured in the different immune cell depleted mice, or that the human brain samples were greater in numbers and included other synucleinopathies than PD (DLB, MSA etc), for comparison, this is not really necessary and would just hold up the publication of important results. It is more important that the current results, which are novel, are made available and that the field as a whole can rapidly begin to build on them and explore additional aspects of the roles of BAMs in aS diseases.

There are some minor areas that can be improved.

The type (location of infection, precise vector, etc) of AAV mediated overexpression should be indicated already on page 5.

The authors should clarify if the phospho-serine 129 aS (pSer) pathology was PK resistant, indicative or true Lewy-like aggregates. Do they have results that address this?

If Fig 1 g is quantification is reporting TH immunostained neurons, it has to be clear in the figure legend. Furthermore, these results should be reported as real numbers, not as percentages of control side.

The authors report no Competing financial interests, but at least one of the authors is a consultant for numerous pharma and biotech companies, and needs to declare this.

Reviewer #3 (Remarks to the Author):

In this study Schonhoff and colleagues investigated the role of border-associated macrophages (BAMs) during neuronal dopaminergic degeneration induced by viral alpha-synuclein expression. Through scRNAseq the authors compared microglia and BAM transcriptional profiling and relative cell composition. Then, after depletion of BAMs with clodronate liposomes, they observed a reduction of CD4, but not CD8, T cells, less monocyte infiltration and mitigated inflammation. In contrast, MHCII silencing in microglia did not prevent T cell and monocyte infiltration. This study provides a first detailed description of this myeloid population in a PD model and highlights its relevant role in modulating the inflammatory response during neurodegeneration. These results are original and cast new light in a poorly characterized myeloid cell population showing its different response respect to microglia to promote the neuroinflammatory response during the neurodegenerative process. Although the results generally support the authors' conclusions some key findings are not fully investigated and require additional experimental support, as in particular:

1) Although the authors showed that depletion of BAMs reduces CD4 T cells and monocyte infiltration, it is not shown if this has an effect in protecting from neurodegeneration. This is a key aspect of the work and the analysis of the relative loss of neuronal dopaminergic cell in this condition should be provided in the revised work.

2) The authors conclude that depletion of BAMs restrains neuroinflammation, but the data related to

this aspect is limited to the apparent reduction of Iba1 immunofluorescence signal in microglial cells (Fig. 5j). Additional markers of the microglial activated state and more objective quantitative methods (i.e. flow-cytometry) should be considered to provide sufficient evidence for this finding. Single-cell or bulk RNAseq of microglia in this condition would be also very informative.

3) The authors showed that depletion of BAMs inhibits the recruitment of CD4, but not CD8, T cells. This is quite unexpected since most of the chemokines stimulate both type of T cells with comparable effects. This finding should be properly elaborated by the authors in the discussion.

4) Double immunofluorescence results on human brain tissues are unfortunately of low quality preventing to clearly highlight T cells within the physical proximity of myeloid cells. In addition, CD68 is a very generic marker of myeloid cells and CD3 labels both CD4 and CD8 T cell populations. This analysis need to be improved including a CD4 T cell specific staining with relative quantifications together with new images where the two cell populations are much better detectable.

5) It is rather surprising that MHCII inactivation in myeloid cells is able alone to completely prevent the loss of dopaminergic neurons. Viral AAV2-Syn expression in neurons is expected to induce neuronal dysfunctions and subsequent death also with a cell-autonomous modality as highlighted in many other studies. The authors are invited to elaborate this aspect with their point-of-view in the discussion.

We thank the reviewers for their detailed feedback on our manuscript. We have addressed the critiques below in detail and have edited the manuscript to reflect these responses.

Reviewer 1:

In their study Schonhoff and colleagues are trying to define a role of CNS macrophages in the interfaces in a PD mouse model. They used several state-of-the art methods such as scRNA-seq profiling as well as Cx3CR1 and TMEM119 Cre models to deplete MHC class II from tissue resident macrophages. They finally conclude that CNS macrophages in the interfaces rather than microglia shape CD4 interactions by regulating antigen presentation in vivo.

Despite the relevance of the topic there are major weaknesses in the study, especially in the conclusion that MHC class II expression by microglia is redundant for disease (see main comments 1). After improving these points the manuscript will be an important for the field.

Main Comments:

1. Fig. 3 is problematic. Since TMEM119-mediated depletion of MHC class II is incomplete they authors can't conclude that antigen expression by microglia is not required in this PD model. This might simply be due to the insufficient gene deletion. To circumvent this problem the authors should definitively use either another microglia-specific Cre line (e.g. Hexb ERT2 Cre) or a BAM/CAM line (Mrc1 ERT2 Cre).

While we agree that the use of additional mouse lines for microglia and/or BAMs would further support our conclusions, unfortunately the suggested mouse lines are not commercially available and their procurement would significantly delay the publication of the manuscript. Furthermore, an exhaustive characterization of microglia-cre drivers from Masuda et al (2020) showed that these lines have comparable recombination efficiencies (using fluorescent reporter mice, the Hexb line has $90.8 \pm 1.6\%$ recombination in microglia whereas our Tmem119 line has 93.4% recombination – see Figure 6f of Masuda et al. (2020) and see Figure 3a and 3c in the current manuscript). For these reasons, we find no reason to believe that the use of additional Cre drivers would substantially change our conclusions. Additionally, our current scRNA sequencing dataset indicates that Hexb expression is found in both BAMs and microglia, albeit at differing levels. See attached u-map below in Figure 1.

Figure 1. Expression of Hexb in CRMs, including BAMs and microglia

Furthermore, and consistent with previously published literature, we have found that Mrc1/CD206 is not specific to BAMs, as monocytes are also capable of expressing it (see Figure 4g in Jordao et al. 2019 in Science). As we know that infiltrating monocytes play integral roles to the neurodegenerative process in the AAV-SYN model, the interpretation of Mrc1 driven recombination would still have substantial limitations. While we agree that specific genetic manipulations would be ideal to provide additional support for our findings, we respectfully disagree that they are necessary given the feasibility and the evidence we have provided within. We have edited the discussion on page 12-13 to reflect these changes.

2. The transcriptional description of BAMs/CAMs in Fig. 2 (scRNA-seq) is unclear. These cells should be characterized by their core genes such as Mrc-1, CD163, Pf4 etc. This has to be shown.

In order to clarify the cluster descriptions of BAMs/CAMs in our scRNA-seq, we have now included a volcano plot highlighting the unique transcriptional profile defining cluster 8 (BAMs) from the remaining clusters (microglia). As seen in Supplemental Figure 2d, enhanced expression of numerous BAM-defining genes such as *Pf4*, *Cd163*, and *Clec12a* was found in cluster 8 while the remaining clusters were found to display high expression of numerous microglial-defining genes including *Tmem119*, *Siglech*, *Hexb*, and *Fcrls*. We thank the reviewer for this opportunity to further display our reasoning in cluster identification.

3. There are no extensive transcriptional differences between BAMs/CAMs. This statement is simply wrong. Very few genes (such as P2yR12, TMEM119, Mrc-1 etc.) are different among the approx. 20,000 genes that a cell has. Please correct this sentence.

We must respectfully disagree with the reviewer on several points in this comment. While we agree that there are ~20,000 genes in the entire genome, the cellular diversity in mammalian systems is due to the unique expression of only ~3000-5000 of these genes. Using a pseudobulk RNA-seq analysis of our scRNA-seq data set showed that of the ~3000 genes transcribed in BAMs and microglia (see Figure 2), over 1000 of them are differentially expressed between the 2 populations (Supplemental Figure 2b). While we agree that the term extensive is vague, we feel that differences in >33% of the cellular transcriptome is significant.

Figure 2. Pseudobulk RNA-seq analysis of cellular clusters of BAMs and microglia. These plots demonstrate the extensive transcriptional difference between populations.

4. Figure 6 is especially important of the study because it shows T cell location next to BAMs/CAMs. However, this figure is problematic by two reasons. First, not enough human samples were assessed. At least 8 samples (patients) per condition should be taken. Second, the shown images lack anatomical information of the midbrain/substantia nigra. Please provide images showing this.

We agree with the reviewer that Figure 6 is important to the translational value of our study. However, we respectfully disagree that a minimum of 8 human samples are required. We have utilized six human samples per group while attaining statistical significance and are unsure how the addition of two extra samples would significantly change the findings. However, we do appreciate the suggestion to include anatomical information clarifying where in the human brain the analysis was performed. An image demonstrating the analyzed regions has been included in Figure 6b.

5. Non-microglia macrophages can be called BAMs or CAMs (CNS-associated macrophages). I personally prefer the latter because it clearly indicates the anatomical location. Please give one citation for the CAM literature Border associated macrophages are present in the whole body.

The reviewer is correct that other literature uses the name CAMs for this cell subset of non-microglial macrophages within the CNS. We have stated this in our manuscript to help dispel confusion and have added a relevant citation for Masuda et al. (2022) in Nature.

6. The sentence on distinct progenitors of BAM/CAM in the yolk sac is wrong. There is a non-committed cell in the YS that can still give rise to both microglia and CAM. Therefore, remove the citation #14 (Utz et al.)

The reviewer is correct, and the sentence has been changed to clarify the early divergence of BAMs and microglia from a common erythro-myeloid progenitor that was demonstrated in Utz et al. 2020 and Masuda et al. 2022

7. Introduction & across the whole manuscript: The term neuroinflammation is used too sloppy. According to a very recent and important consensus paper in microglial nomenclature in Neuron 2022 by Paolicelli and many others (please cite this work), this term should be reserved for conditions of robust influx of circulating cells such as MS, stroke etc. rather than for classical neurodegenerative conditions such as AD, PD, ALS. I truly believe that T cell infiltrates in mouse models of neuroinflammation are highly artificial.

We have edited the manuscript to be selective with our use of the term neuroinflammation. We have also added the citation as suggested. While we respect the reviewer's opinion on the importance of adaptive immune responses in neurodegeneration, we would like to highlight the extensive evidence for T cell infiltration in the CNS in both human disease and animal models, evidence for alpha-synuclein reactive T cells circulating in the periphery in human disease, and the strong genetic association between PD and MHCII (the antigen presentation complex used exclusively for CD4 T-cells). We thank the reviewer for the opportunity to discuss differing viewpoints.

8. Fig. 1 D: white balance is needed for these images. Moreover, the magnification of the images is suboptimal.

We have edited the images and provided higher magnification for clarity.

9. There is no evidence for different populations of BAMs rather than different clusters representing different states. Please reword these sentences.

We have re-worded these sentences on page 8 to reflect the unknown nature of heterogeneity within the BAM population.

Reviewer #2

This paper describes for the first time the role of Border Associated Macrophages (BAMs) in the context of a Parkinson disease (PD) model which involves overexpression of the disease-associated (and aggregation prone) protein alpha-synuclein (aS).

The paper describes an extensive and important set of experiments which include aS overexpression, selective depletion of sub populations of immune cells, neuropathology, single cell RNAseq data. Most experiments are in mouse models but the study has incorporated human brain tissue samples. It is well written and the results are well documented. Because the role of BAMs in PD is not clearly defined, there is a significant impact on the field. Overall, studies that demonstrate a role for the immune system in PD are important because they provide fundamental insights into disease mechanisms and the origins of the disease, as well as can highlight novel potential therapeutic targets. Notably, the authors suggest that microglia are not the primary antigen presenting cells in this context, but that BAMs subserve this role. This is a key, potentially paradigm-shifting, finding.

While it can be tempting for a referee to suggest that cytokines were measured in the different immune cell depleted mice, or that the human brain samples were greater in numbers and included other synucleinopathies than PD (DLB, MSA etc), for comparison, this is not really necessary and would just hold up the publication of important results. It is more important that the current results, which are novel, are made available and that the field as a whole can rapidly begin to build on them and explore additional aspects of the roles of BAMs in aS diseases.

1. The type (location of infection, precise vector, etc) of AAV mediated overexpression should be indicated already on page 5.

We have added this information on page 5, as suggested.

2. The authors should clarify if the phospho-serine 129 aS (pSer) pathology was PK resistant, indicative of true Lewy-like aggregates. Do they have results that address this?

The nature of the α -syn in this model has been addressed in a previous publication from our lab. Previous validation of this model indicated the α -syn in the SNpc is triton insoluble. We have added this important information into the manuscript and included the relevant citation on page 5.

3. If Fig 1 g is quantification is reporting TH immunostained neurons, it has to be clear in the figure legend. Furthermore, these results should be reported as real numbers, not as percentages of control side

We thank the reviewer for this suggestion and have clarified this information in the figure legend to indicate that Figure 1g represents the quantification of TH+ neurons. Additionally, we have included raw neuron counts in Supplemental Figure 1c for added transparency.

4. The authors report no Competing financial interests, but at least one of the authors is a consultant for numerous pharma and biotech companies, and needs to declare this.

While none of the authors have competing financial interests for the data included within the manuscript, we have updated the competing interests section with their other disclosures for increased transparency.

Reviewer #3

In this study Schonhoff and colleagues investigated the role of border-associated macrophages (BAMs) during neuronal dopaminergic degeneration induced by viral alpha-synuclein expression. Through scRNAseq the authors compared microglia and BAM transcriptional profiling and relative cell composition. Then, after depletion of BAMs with clodronate liposomes, they observed a reduction of CD4, but not CD8, T cells, less monocyte infiltration and mitigated inflammation. In contrast, MHCII silencing in microglia did not prevent T cell and monocyte infiltration. This study provides a first detailed description of this myeloid population in a PD model and highlights its relevant role in modulating the inflammatory response during neurodegeneration. These results are original and cast new light in a poorly characterized myeloid cell population showing its different response respect to microglia to promote the neuroinflammatory response during the neurodegenerative process. Although the results generally support the authors' conclusions some key findings are not fully investigated and require additional experimental support, as in particular:

1. Although the authors showed that depletion of BAMS reduces CD4 T cells and monocyte infiltration, it is not shown if this has an effect in protecting from neurodegeneration. This is a key aspect of the work and the analysis of the relative loss of neuronal dopaminergic cell in this condition should be provided in the revised work.

Figure 3. Quantification of BAMS in the ventral midbrain one month post injection with SL or CL.

We agree that further demonstration of the connection between BAMS and neurodegeneration is an important future direction. However, due to technical and animal welfare limitations we have been unable to fully confirm these findings. The clodronate liposomes used to deplete BAMS provide temporary depletion, allowing for experiments with shorter timepoints. However, a substantial limitation to these experiments is the limited time frame of clodronate's efficacy. Preliminary control experiments in our lab found that BAMS had repopulated the CNS

by approximately one month post-depletion, and it was for this reason that we selected the 4 week timepoint (Figure 3). Because this animal model requires ~6 months to show neurodegeneration of TH+ cells and due to concerns over animal welfare raised by Institutional Animal Care and Use Committee over repeated i.c.v. injections, we are unable to perform this long-term experiment. We do recognize this is a current limitation of our study and we have updated the discussion on page 12-13 to address this.

2. The authors conclude that depletion of BAMS restrains neuroinflammation, but the data related to this aspect is limited to the apparent reduction of Iba1 immunofluorescence signal in microglial cells (Fig. 5j). Additional markers of the microglial activated state and more objective quantitative methods (i.e. flow-cytometry) should be considered to provide sufficient evidence for this finding. Single-cell or bulk RNAseq of microglia in this condition would be also very informative.

To address this comment, flow cytometry experiments displayed in Figure 5 indicate that BAM depletion attenuates microglial upregulation of MHCII, infiltration of Ly6C^{hi} monocytes, and CD4+ T cells. Additionally, we have confocal images labelled for MHCII and IBA1 demonstrating the decreased inflammatory response in the SNpc. We have also added flow cytometric experiments in Supplemental Figure 2 to better characterize the microglial immune response to a-syn overexpression. We have included quantification of microglial PD-L1, Arg1, iNOS, and CD68 expression.

3. The authors showed that depletion of BAMS inhibits the recruitment of CD4, but not CD8, T cells. This is quite unexpected since most of the chemokines stimulate both type of T cells with comparable effects. This finding should be properly elaborated by the authors in the discussion.

We agree with the reviewer and have added a brief discussion of our findings in the discussion section on page 15.

4. Double immunofluorescence results on human brain tissues are unfortunately of low quality preventing to clearly highlight T cells within the physical proximity of myeloid cells. In addition, CD68 is a very generic marker of myeloid cells and CD3 labels both CD4 and

CD8 T cell populations. This analysis need to be improved including a CD4 T cell specific staining with relative quantifications together with new images where the two cell populations are much better detectable.

We have added additional images of both CD4+ and CD8+ T cells in the perivascular space within the nigra of PD postmortem brain. Intriguingly, many of the cells in the perivascular space in postmortem tissue were CD8+ T cells, a finding which is unsurprising given the current literature that CD4 T cells are active early in PD, while CD8 T cells are present in later disease stages. We have addressed this result in the discussion section on page 14. Furthermore, we have edited images to highlight the difference between CD68+ BAMs and CD3+ T cells.

5. It is rather surprising that MHCII inactivation in myeloid cells is able alone to completely prevent the loss of dopaminergic neurons. Viral AAV2-Syn expression in neurons is expected to induce neuronal dysfunctions and subsequent death also with a cell-autonomous modality as highlighted in many other studies. The authors are invited to elaborate this aspect with their point-of-view in the discussion.

Previously, our lab has demonstrated that global knock-out of MHCII is sufficient to prevent neurodegeneration of TH+ neurons in the SNpc. While we have not assayed the absence or presence of neuronal dysfunction in these manuscripts, previous publications utilizing the same AAV2-SYN vector were able to detect dystrophic neurites in the SNpc indicating neuronal dysfunction (See Theodore et al. 2008 in J Neuropathol Exp Neurol., Cao et al. 2010 in Molecular Neurodegeneration, and St. Martin et al. 2997 in Journal of Neurochemistry). We hypothesize that the absence of an immune response could lead to a slower or delayed disease course. We have updated the discussion on page 14 to reflect these points.

REVIEWERS' COMMENTS

Reviewer #1 (Remarks to the Author):

The authors addressed all my concerns and provided a nice revision of the study. I have no further comments.

Reviewer #2 (Remarks to the Author):

The authors have responded to my concerns in a satisfactory manner. I have no further comments at this point.

Reviewer #3 (Remarks to the Author):

The authors performed additional work in order to properly elaborate the comments raised by this reviewer. However, they were unable to solve the key point raised in the first submission, that is whether depletion of BAMs is sufficient to rescue the degeneration of dopaminergic neurons in their mouse model. The authors mentioned that clodronate liposomes have short-term effects, but it would be expected that they could come with additional methods to perform long-term depletion of BAMs to properly assess this critical aspect. In fact, at this point it remains unsolved if limiting BAM accumulation can lead ultimately to rescue or delay neuronal degeneration progression. As an incremental point, the authors show that partial loss of BAMs by clodronate liposomes induces only a roughly 50% reduction of CD4+ T cells and monocytes. Thus, in these conditions it remains unsolved if this partial effect is caused by the inefficient approach or due to other myeloid populations that have a substantial role in this process. Again, a more efficient system to deplete BAMs would be very welcome to provide more convincing evidence to solve this legitimate doubt. Since this is the cardinal aspect of the work and the conclusions are based on these last results, a more definitive reduction of BAMs and its consequences would be essential to add to this study.

We thank the reviewers for their detailed feedback on our manuscript. We have addressed the critiques below in detail and have edited the manuscript to reflect these responses.

Reviewer 1:

Main Comments:

1. Fig. 3 is problematic. Since TMEM119-mediated depletion of MHC class II is incomplete they authors can't conclude that antigen expression by microglia is not required in this PD model. This might simply be due to the insufficient gene deletion. To circumvent this problem the authors should definitively use either another microglia-specific Cre line (e.g. Hexb ERT2 Cre) or a BAM/CAM line (Mrc1 ERT2 Cre).

While we agree that the use of additional mouse lines for microglia and/or BAMs would further support our conclusions, unfortunately the suggested mouse lines are not commercially available and their procurement would significantly delay the publication of the manuscript. Furthermore, an exhaustive characterization of microglia-cre drivers from Masuda et al (2020) showed that these lines have comparable recombination efficiencies (using fluorescent reporter mice, the Hexb line has $90.8 \pm 1.6\%$ recombination in microglia whereas our Tmem119 line has 93.4% recombination – see Figure 6f of Masuda et al. (2020) and see Figure 3a and 3c in the current manuscript). For these reasons, we find no reason to believe that the use of additional Cre drivers would substantially change our conclusions. Additionally, our current scRNA sequencing dataset indicates that Hexb expression is found in both BAMs and microglia, albeit at differing levels. See attached u-map below in Figure 1.

Figure 1. Expression of *Hexb* in CRMs, including BAMs and microglia

Furthermore, and consistent with previously published literature, we have found that Mrc1/CD206 is not specific to BAMs, as monocytes are also capable of expressing it (see Figure 4g in Jordao et al. 2019 in Science). As we know that infiltrating monocytes play integral roles to the neurodegenerative process in the AAV-SYN model, the interpretation of Mrc1 driven recombination would still have substantial limitations. While we agree that specific genetic manipulations would be ideal to provide additional support for our findings, we respectfully disagree that they are necessary given the feasibility and the evidence we have provided within. We have edited the discussion on page 12-13 to reflect these changes.

2. The transcriptional description of BAMs/CAMs in Fig. 2 (scRNA-seq) is unclear. These cells should be characterized by their core genes such as Mrc-1, CD163, Pf4 etc. This has to be shown.

In order to clarify the cluster descriptions of BAMs/CAMs in our scRNA-seq, we have now included a volcano plot highlighting the unique transcriptional profile defining cluster 8 (BAMs) from the remaining clusters (microglia). As seen in Supplemental Figure 2d, enhanced expression of numerous BAM-defining genes such as Pf4, Cd163, and Clec12a was found in cluster 8 while the remaining clusters were found to display high expression of numerous microglial-defining genes including Tmem119, Siglech, Hexb, and Fcrls. We thank the reviewer for this opportunity to further display our reasoning in cluster identification.

3. There are no extensive transcriptional differences between BAMs/CAMs. This statement is simply wrong. Very few genes (such as P2yR12, TMEM119, Mrc-1 etc.) are different among the approx. 20,000 genes that a cell has. Please correct this sentence.

We must respectfully disagree with the reviewer on several points in this comment. While we agree that there are ~20,000 genes in the entire genome, the cellular diversity in mammalian systems is due to the unique expression of only ~3000-5000 of these genes. Using a pseudobulk RNA-seq analysis of our scRNA-seq data set showed that of the ~3000 genes transcribed in BAMs and microglia (see Figure 2), over 1000 of them are differentially expressed between the 2 populations (Supplemental Figure 2b). While we agree that the term extensive is vague, we feel that differences in >33% of the cellular transcriptome is significant.

Figure 2. Pseudobulk RNA-seq analysis of cellular clusters of BAMs and microglia. These plots demonstrate the extensive transcriptional difference between populations.

4. Figure 6 is especially important of the study because it shows T cell location next to BAMs/CAMs. However, this figure is problematic by two reasons. First, not enough human samples were assessed. At least 8 samples (patients) per condition should be taken. Second, the shown images lack anatomical information of the midbrain/substantia nigra. Please provide images showing this.

We agree with the reviewer that Figure 6 is important to the translational value of our study. However, we respectfully disagree that a minimum of 8 human samples are required. We have utilized six human samples per group while attaining statistical significance and are unsure how the addition of two extra samples would significantly change the findings. However, we do appreciate the suggestion to include anatomical information clarifying where in the human brain the analysis was performed. An image demonstrating the analyzed regions has been included in Figure 6b.

5. Non-microglia macrophages can be called BAMs or CAMs (CNS-associated macrophages). I personally prefer the latter because it clearly indicates the anatomical location. Please give one citation for the CAM literature. Border associated macrophages are present in the whole body.

The reviewer is correct that other literature uses the name CAMs for this cell subset of non-microglial macrophages within the CNS. We have stated this in our manuscript to help dispel confusion and have added a relevant citation for Masuda et al. (2022) in Nature.

6. The sentence on distinct progenitors of BAM/CAM in the yolk sac is wrong. There is a non-committed cell in the YS that can still give rise to both microglia and CAM. Therefore, remove the citation #14 (Utz et al.)

The reviewer is correct, and the sentence has been changed to clarify the early divergence of BAMs and microglia from a common erythro-myeloid progenitor that was demonstrated in Utz et al. 2020 and Masuda et al. 2022

7. Introduction & across the whole manuscript: The term neuroinflammation is used too sloppy. According to a very recent and important consensus paper in microglial nomenclature in Neuron 2022 by Paolicelli and many others (please cite this work), this term should be reserved for conditions of robust influx of circulating cells such as MS, stroke etc. rather than for classical neurodegenerative conditions such as AD, PD, ALS. I truly believe that T cell infiltrates in mouse models of neuroinflammation are highly artificial.

We have edited the manuscript to be selective with our use of the term neuroinflammation. We have also added the citation as suggested. While we respect the reviewer's opinion on the importance of adaptive immune responses in neurodegeneration, we would like to highlight the extensive evidence for T cell infiltration in the CNS in both human disease and animal models, evidence for alpha-synuclein reactive T cells circulating in the periphery in human disease, and the strong genetic association between PD and MHCII (the antigen presentation complex used exclusively for CD4 T-cells). We thank the reviewer for the opportunity to discuss differing viewpoints.

8. Fig. 1 D: white balance is needed for these images. Moreover, the magnification of the images is suboptimal.

We have edited the images and provided higher magnification for clarity.

9. There is no evidence for different populations of BAMs rather than different clusters representing different states. Please reword these sentences.

We have re-worded these sentences on page 8 to reflect the unknown nature of heterogeneity within the BAM population.

Reviewer #2

1. The type (location of infection, precise vector, etc) of AAV mediated overexpression should be indicated already on page 5.

We have added this information on page 5, as suggested.

2. The authors should clarify if the phospho-serine 129 aS (pSer) pathology was PK resistant, indicative of true Lewy-like aggregates. Do they have results that address this?

The nature of the α -syn in this model has been addressed in a previous publication from our lab. Previous validation of this model indicated the α -syn in the SNpc is triton insoluble. We have added this important information into the manuscript and included the relevant citation on page 5.

3. If Fig 1 g is quantification is reporting TH immunostained neurons, it has to be clear in the figure legend. Furthermore, these results should be reported as real numbers, not as percentages of control side

We thank the reviewer for this suggestion and have clarified this information in the figure legend to indicate that Figure 1g represents the quantification of TH+ neurons. Additionally, we have included raw neuron counts in Supplemental Figure 1c for added transparency.

4. The authors report no Competing financial interests, but at least one of the authors is a consultant for numerous pharma and biotech companies, and needs to declare this.

While none of the authors have competing financial interests for the data included within the manuscript, we have updated the competing interests section with their other disclosures for increased transparency.

Reviewer #3

1. Although the authors showed that depletion of BAMS reduces CD4 T cells and monocyte infiltration, it is not shown if this has an effect in protecting from neurodegeneration. This is a key aspect of the work and the analysis of the relative loss of neuronal dopaminergic cell in this condition should be provided in the revised work.

Figure 3. Quantification of BAMS in the ventral midbrain one month post injection with SL or CL.

by approximately one month post-depletion, and it was for this reason that we selected the 4 week timepoint (Figure 3). Because this animal model requires ~6 months to show neurodegeneration of TH+ cells and due to concerns over animal welfare raised by Institutional Animal Care and Use Committee over repeated i.c.v. injections, we are unable to perform this long-term experiment. We do recognize this is a current limitation of our study and we have updated the discussion on page 12-13 to address this.

We agree that further demonstration of the connection between BAMS and neurodegeneration is an important future direction. However, due to technical and animal welfare limitations we have been unable to full confirm these findings. The clodronate liposomes used to deplete BAMS provide temporary depletion, allowing for experiments with shorter timepoints. However, a substantial limitation to these experiments is the limited time frame of clodronate's efficacy. Preliminary control experiments in our lab found that BAMS had repopulated the CNS

2. The authors conclude that depletion of BAMS restrains neuroinflammation, but the data related to this aspect is limited to the apparent reduction of Iba1 immunofluorescence signal in microglial cells (Fig. 5j). Additional markers of the microglial activated state and more objective quantitative methods (i.e. flow-cytometry) should be considered to provide sufficient evidence for this finding. Single-cell or bulk RNAseq of microglia in this condition would be also very informative.

To address this comment, flow cytometry experiments displayed in Figure 5 indicate that BAM depletion attenuates microglial upregulation of MHCII, infiltration of Ly6C^{hi} monocytes, and CD4+ T cells. Additionally, we have confocal images labelled for MHCII and IBA1 demonstrating the decreased inflammatory response in the SNpc. We have also added flow cytometric experiments in Supplemental Figure 2 to better characterize the microglial immune response to α -syn overexpression. We have included quantification of microglial PD-L1, Arg1, iNOS, and CD68 expression.

3. The authors showed that depletion of BAMS inhibits the recruitment of CD4, but not CD8, T cells. This is quite unexpected since most of the chemokines stimulate both type of T cells with comparable effects. This finding should be properly elaborated by the authors in the discussion.

We agree with the reviewer and have added a brief discussion of our findings in the discussion section on page 15.

4. Double immunofluorescence results on human brain tissues are unfortunately of low quality preventing to clearly highlight T cells within the physical proximity of myeloid cells. In addition, CD68 is a very generic marker of myeloid cells and CD3 labels both CD4 and CD8 T cell populations. This analysis need to be improved including a CD4 T cell specific staining with relative quantifications together with new images where the two cell populations are much better detectable.

We have added additional images of both CD4+ and CD8+ T cells in the perivascular space within the nigra of PD postmortem brain. Intriguingly, many of the cells in the perivascular space in postmortem tissue were CD8+ T cells, a finding which is unsurprising given the current literature that CD4 T cells are active early in PD, while CD8 T cells are present in later disease stages. We have addressed this result in the discussion section on page 14. Furthermore, we have edited images to highlight the difference between CD68+ BAMs and CD3+ T cells.

5. It is rather surprising that MHCII inactivation in myeloid cells is able alone to completely prevent the loss of dopaminergic neurons. Viral AAV2-Syn expression in neurons is expected to induce neuronal dysfunctions and subsequent death also with a cell-autonomous modality as highlighted in many other studies. The authors are invited to elaborate this aspect with their point-of-view in the discussion.

Previously, our lab has demonstrated that global knock-out of MHCII is sufficient to prevent neurodegeneration of TH+ neurons in the SNpc. While we have not assayed the absence or presence of neuronal dysfunction in these manuscripts, previous publications utilizing the same AAV2-SYN vector were able to detect dystrophic neurites in the SNpc indicating neuronal dysfunction (See Theodore et al. 2008 in J Neuropathol Exp Neurol., Cao et al. 2010 in Molecular Neurodegeneration, and St. Martin et al. 2997 in Journal of Neurochemistry). We hypothesize that the absence of an immune response could lead to a slower or delayed disease course. We have updated the discussion on page 14 to reflect these points.